# Loss of *Bcl6* promotes antitumor immunity by activating glycolysis to rescue CD8 T-cell function

Fangkun Luan[1,2,*], Yunqiao Li[2,3,*], Jia Ning[4], Jenny Tuyet Tran[2], Tanya R Blane[2], Raag Bhargava[2], Zhe Huang[2], Changchun Xiao[2], David Nemazee[2]

T cells are one of the most powerful weapons to fight cancer; however, T-cell exhaustion and dysfunction restrict their long-lasting function in antitumor immunity. B-cell lymphoma 6 (BCL6) has many functions in CD8 T cells; however, it is unclear how it regulates the effector function and exhaustion of CD8 cells. Overall, a low level of BCL6 mRNA in human cancer samples is associated with better outcomes, but high expression of BCL6 is specifically observed in cytotoxic CD8 T cells. We found that BCL6 deficiency in activated CD8 T cells enhanced tumor repression in multiple mouse models. More IL-2–expressing CD8 T cells and reduced proportions of exhausted or dysfunctional CD8 T cells were detected within tumors when *Bcl6* was knocked out upon T-cell activation. Glycolysis was promoted, and GLUT3 expression was derepressed in BCL6-deficient CD8 T cells. The BCL6 inhibitor Fx1 promoted antitumor immunity in a T cell–dependent manner. These findings suggest a novel pathway to restore effector function of CD8 T cells by changing their energy use pathways to facilitate long-term tumor resistance.

## Introduction

Upon T-cell activation, naïve CD8 T cells differentiate into either cytotoxic effector or memory T cells. Effector T cells rapidly proliferate and secrete killing cytokines to help with pathogen clearance. After the acute response, only a small number of effector T cells survive and develop into memory cells to provide long-term immune protection. When T cells are exposed to persistent antigen stimulation, such as chronic viral infection or tumors, activated T cells become exhausted or dysfunctional, losing their effector functions. The category of exhausted T cells encompasses a broad definition, characterized by persistently high expression of inhibitory receptors, such as PD1, TIM3, LAG3, CTLA4,

and TIGIT, reduced proliferative ability upon stimulation, decreased effector cytokine secretion, and a unique epigenetic profile (1, 2). Exhausted T cells appear to lose their ability to kill target cells and occupy a stable T differentiation state (3) distinct from memory T cells. Moreover, the exhausted T-cell population is heterogeneous, containing at least progenitor and terminal subsets (4). Progenitor-exhausted T cells have high levels of expression of TCF1, SLAMF6, and CXCR5 (5, 6, 7, 8), whereas cells expressing TIM-3 and CD39 identify terminally exhausted T cells (9, 10). Markers Ly108 (encoded by *Slamf6*) and CD69 can further divide four subsets among those exhausted CD8 T cells (11). In addition to these characteristics of exhausted T cells, their energy metabolism is different from effector or memory T cells. Effector T cells rely on aerobic glycolysis for rapid proliferation and effector activities, whereas memory T cells perform elevated oxidative phosphorylation (OXPHOS) and autophagy to maintain their long-term survival and function (10, 12, 13, 14, 15). In contrast to the energy requirements of effector and memory T cells, exhausted T cells demonstrate bioenergetic insufficiencies, such as suppressed mitochondrial respiration and poor glycolysis (16, 17, 18). It is important to investigate ways to restore effector functions in exhausted T cells to facilitate an antitumor response.

BCL6 is a transcriptional repressor reported to participate in many types of regulation of T and B cells (19). BCL6 binds to the promoter region of *Gzmb* (encoding granzyme B) to repress its expression in naïve CD8 T cells rather than in activated CD8 T cells (20). BCL6 is also critical for maintaining and generating memory CD8 T cells (21). The level of BCL6 expression in T cells after antigen stimulation positively correlates with their later differentiation into central memory T cells and antigen-specific TCM secondary expansion (22). As a well-established antagonist of BLIMP1 activity, BCL6 promotes formation of stem-like CD8[+] T cells with memory potential and high proliferation capacity (23). In acute viral infection, BCL6 promotes the expression of TCF-1 to generate CD8 memory precursors but is not required to maintain memory CD8 T cells (24). Because memory T cells or exhausted T cells are generated from effectors, it is unclear whether BCL6 has any role in

---

[1]The School of Medicine, Nankai University, Tianjin, China    [2]Department of Immunology and Microbiology, The Scripps Research Institute, La Jolla, CA, USA    [3]Department of Immunology, Suzhou Medical College of Soochow University, Suzhou, China    [4]Molecular and Cell Biology Laboratory, The Salk Institute, La Jolla, CA, USA

Correspondence: nemazee@scripps.edu; yunli@scripps.edu; cxiao@scripps.edu
*Fangkun Luan and Yunqiao Li contributed equally to this work

---

the differentiation of T-cell exhaustion, its functional window period, or whether it can be used as a target of cancer treatment.

# Results

## Conditional knockout of *Bcl6* in activated CD8 T cells represses tumor growth

To investigate the association between *Bcl6* expression and the outcomes of tumor samples, we took advantage of the GEPIA database (25), which can be used to analyze the RNA-sequencing data of tumor samples under the TCGA and GTEx projects. We found that tumor cases with lower levels of *Bcl6* mRNA had favorable overall survival rates in all types of cancer (Fig S1A) and this phenotype was confirmed in colon adenocarcinoma (COAD, with *P* = 0.049), pancreatic adenocarcinoma (PAAD, with *P* = 0.021), and, possibly, rectum adenocarcinoma (READ, no significant difference but trend was shown) (Fig S1B–D). Moreover, tumor-infiltrating CD8 T-cell levels and *Bcl6* expression showed a positive correlation in COAD, PAAD, and READ samples (Fig S1E–G), suggesting that tumor-intrinsic BCL6 expression is associated with a more aggressive malignant phenotype, even in the presence of increased T-cell infiltration. However, when BCL6 levels in CD8 T cells were analyzed across scRNA-seq datasets from colon cancer samples, we found cytotoxic or effector CD8 T cells had higher expression of BCL6 (Figs 1A and S1H), which is encouraging for studying the distinct gene suppressive function of BCL6 in T-cell responses during cancer.

To study the function of BCL6 in the tumor immune response, we analyzed tumor growth in various conditionally deficient BCL6 models. *Bcl6*<sup>fl/fl</sup> CD4-cre mice, which lack BCL6 in both CD4 and CD8 T cells, starting at the double-positive stage of T-cell development, showed inhibited growth of MC38, LLC1 (Lewis lung carcinoma), and B16F10 (melanoma) tumors compared with *Bcl6*<sup>fl/fl</sup> control mice (Fig S2A–G). This contribution to tumor resistance likely did not come from CD4 T cells as *Bcl6*<sup>fl/fl</sup> OX40-cre mice showed tumor growth curves comparable to *Bcl6*<sup>fl/fl</sup> controls (Fig S2H). In contrast, *Bcl6*<sup>fl/fl</sup> Gzmb-cre mice showed repressed growth of MC38, LLC1, and B16F10 tumors, suggesting that the lack of BCL6 in activated CD8 T cells was crucial for enhanced antitumor immunity (Fig 1B–F). We validated that in *Bcl6*<sup>fl/fl</sup> Gzmb-cre mice, BCL6 deficiency occurred in activated CD8 T cells rather than in CD4 T cells (Fig S2I).

## Lack of BCL6 retards exhaustion of CD8⁺ T cells, restoring their effector functions

The data described above, indicating that repressing BCL6 expression in activated CD8 T cells can benefit tumor control and may result in changes in CD8 T-cell effector function, encouraged us to compare MC38 tumor-infiltrating CD8 cells in *Bcl6*<sup>fl/fl</sup> and *Bcl6*<sup>fl/fl</sup> Gzmb-cre mice. On day 8 post-inoculation (DPI 8) of MC38, *Bcl6*<sup>fl/fl</sup> Gzmb-cre mice had more tumor-infiltrating, effector IFNγ+ CD8 T cells compared with *Bcl6*<sup>fl/fl</sup> mice, though significantly reduced proportions of IFNγ+ CD8 T cells were observed in both groups by DPI 20 (Fig 2A). However,

the proportions of tumor-infiltrating cells producing IL-2 increased in BCL6-deficient CD8 T cells on both DPI 8 and DPI 20 (Fig 2B). There was a small, but statistically insignificant, decrease in TCF1 expression when comparing activated CD44⁺ CD8 T cells of *Bcl6*<sup>fl/fl</sup> Gzmb-cre to *Bcl6*<sup>fl/fl</sup> mice on DPI 8 (Fig 2C), and the number and percentage of tumor-infiltrating TCF1+CD44⁺ CD8 T cells between these two groups on DPI 8 were also comparable (Fig 2D), suggesting that knocking out BCL6 during CD8 T-cell activation may slightly influence their stemness and memory differentiation. On the other hand, the numbers of tumor-infiltrating "exhausted" TIM3+PD1+ CD8 T cells were significantly reduced in *Bcl6*<sup>fl/fl</sup> Gzmb-cre mice on DPI 20 (Fig 2E). Moreover, the CD39+PD1+ dysfunctional CD8 T cells, which were obviously increased in proportion within tumors from DPI 8 to DPI 20 in control mice, were significantly reduced on DPI 20 in *Bcl6*<sup>fl/fl</sup> Gzmb-cre mice (Fig 2F). Consistently, both CD39⁺CD69+PD1+ and Tim-3+PD1+CD39⁺CD69⁺ CD8 T-cell populations within tumors were found significantly reduced in *Bcl6*<sup>fl/fl</sup> Gzmb-cre mice on DPI 20 (Fig 2G and H). These data suggest that the elimination of BCL6 in activated CD8 T cells can prevent the acquisition of features of T-cell exhaustion within tumors and may allow them to exert their killing effector for a longer time.

## Glycolysis is improved in BCL6-deficient CD8 T cells

To investigate how BCL6 regulates activated CD8 T-cell function, we performed bulk RNA-seq, comparing sorted living PD1+CD8 T cells isolated from DPI 10 tumors of MC38-inoculated *Bcl6*<sup>fl/fl</sup> Gzmb-cre (BCL6-KO) with those of *Bcl6*<sup>fl/fl</sup> (BCL6-WT) mice. enrichKEGG analysis of the up-regulated genes indicated that cell cycle, biosynthesis, and energy metabolism regulation were stimulated in BCL6-KO CD8 T cells (Fig 3A, Table S1). Consistently, the KEGG enrichment network showed improvement of glycolysis when BCL6 expression was restricted (Fig 3B, Table S1). Genes associated with glycolysis were increased in activated BCL6-KO CD8 T cells (Fig 3C). To test glycolysis function, we performed metabolic stress tests using naïve or TCR-stimulated CD8 T cells from *Bcl6*<sup>fl/fl</sup> CD4-cre mice. Basal glycolysis in BCL6-KO CD8 T cells was significantly higher than in BCL6-WT CD8 T cells, and the activated BCL6-KO CD8 T cells had better glycolytic compensation when mitochondrial respiration was inhibited (Fig 3D). Glucose uptake of CD8 T cells was detected with or without CD3/CD28 stimulation conditions; notably, BCL6-KO CD8 T cells had significantly enhanced glucose uptake upon TCR stimulation compared with BCL6-WT CD8 T cells (Fig 3E). Higher levels of glucose transporter GLUT3 were observed in both resting and activated BCL6-KO CD8 T cells (Fig 3F), consistent with their stronger glucose uptake. Increased expression of GLUT3 was also detected on tumor-infiltrating CD8+PD1+ T cells of *Bcl6*<sup>fl/fl</sup> Gzmb-cre mice on DPI 20 (Fig 3G). These data demonstrate enhanced glycolysis in BCL6-deficient CD8 T cells at least through up-regulation of glucose transporter GLUT3 expression and GLUT3-mediated glucose uptake.

## BCL6 binds the promoter region of *Slc2a3* and represses its expression

GLUT3 is encoded by *Slc2a3*. We performed a CUT&RUN experiment to quantitate immunoprecipitation of DNA segments bound by

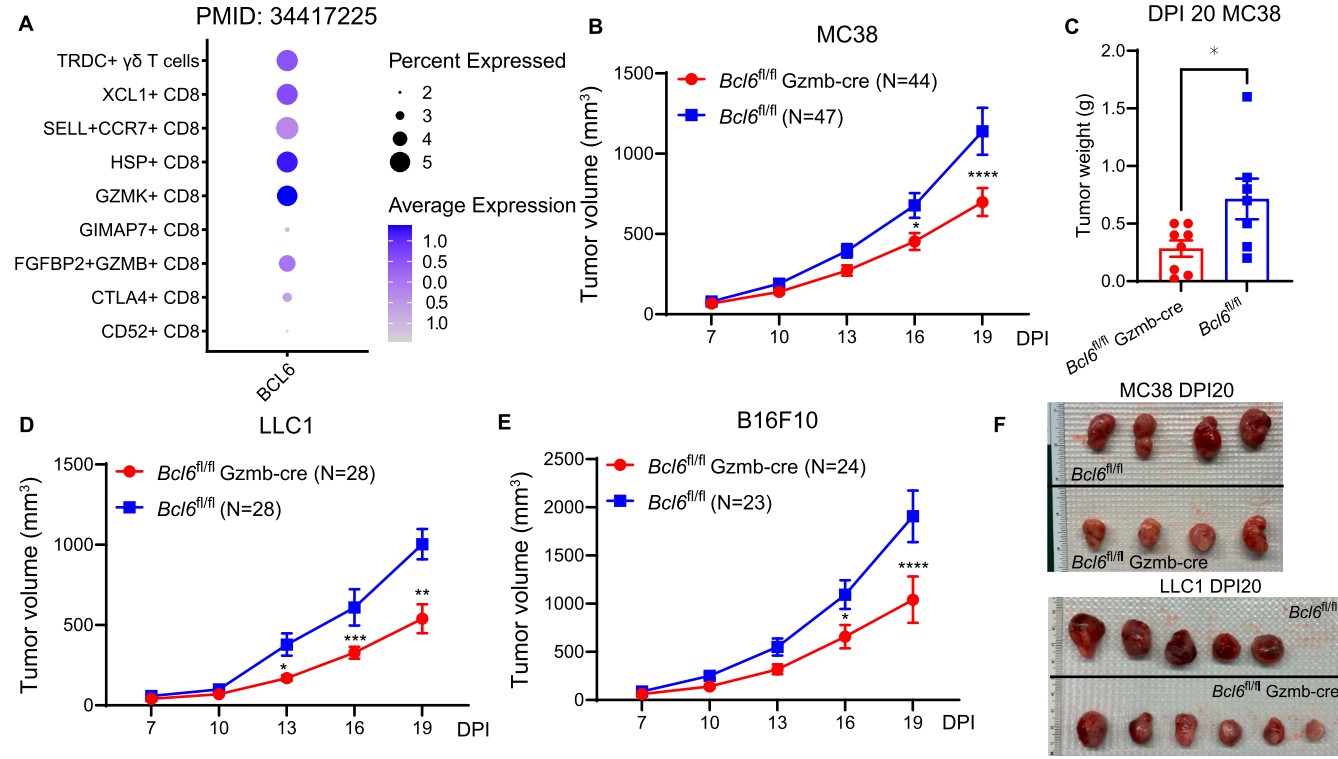

**Figure 1. Low level of BCL6 is correlated with improved survival and reduced tumor growth.**
**(A)** Dot plot showing the expression of BCL6 in different subclusters of CD8 T cells in colorectal cancer liver metastasis samples (26). **(B, C)** $Bcl6^{fl/fl}$ and $Bcl6^{fl/fl}$ GzmB-cre mice were inoculated with 1 × $10^6$ MC38 tumor cells on day 0. **(B)** Tumors were measured every 3 d from day 7 post-inoculation (DPI) revealing slower tumor growth in $Bcl6^{fl/fl}$ GzmB-cre mice (B). **(C)** Comparison of tumor weights on DPI 20 (C). **(D, E)** Similar tumor repression phenotypes were observed in $Bcl6^{fl/fl}$ GzmB-cre mice against LLC1 (D) and B16F10 (E) tumor cell inoculations. **(F)** Visual analysis of tumors of MC38 (upper) and LLC1 (lower) on DPI 20 comparing $Bcl6^{fl/fl}$ GzmB-cre with $Bcl6^{fl/fl}$ tumor recipients. Tumor growth data were pooled from at least four independent experiments. **(C)** Each point in graph (C) represents the value obtained in an independent mouse. Bars are presented as the mean ± SEM. **(B, C, D, E)** P-values were calculated with the two-tailed unpaired t test (C) or two-way ANOVA (B, D, E). * denotes $P < 0.05$.

transcription factors. Compared with the IgG control group, BCL6 was significantly enriched in the promoter regions of $Slc2a3$ (Fig 3H). Luciferase assay showed $Slc2a3$ promoter–driven reporter (Luc2) gene expression was repressed by full-length BCL6, whereas the suppression of Luc2 expression was recovered when BCL6 lost its DNA binding ability (deletion of ZF fragment, ΔZF (27)) (Fig 3I). These data indicate that GLUT3 expression is repressed by BCL6 binding to its promoter regions.

### Fx1, a BCL6 inhibitor, suppresses tumors in a T cell–dependent manner

To determine whether CD8 T-cell exhaustion could be treated pharmacologically, we decided to test the antitumor efficacy and mechanism of action of the BCL6 inhibitor Fx1. We first confirmed that BCL6 expression was strongly induced upon T-cell activation (Fig 4A). Fx1 treatment, given by daily intraperitoneal (i.p.) injection starting on day 10 after MC38 or B16F10 tumor inoculation, repressed tumor growth in C57BL/6 mice (Fig 4B and C). As BCL6 is a master regulator of the cell cycle, we considered the possibility that Fx1 may work to inhibit tumor expansion directly. This seemed unlikely, however, as the antitumor efficacy was lost when the therapy was introduced in $Rag1^{-/-}$ mice, which lack T and B cells (Fig 4B and C). As $Rag1^{-/-}$ mice are sufficient in NK cells, the result

also indicated that inhibiting BCL6 in NK cells was not as important as targeting BCL6 in T cells or B cells. Moreover, as MC38 and B16F10 growth curves were comparable in $Bcl6^{fl/fl}$ Mb1-cre and $Bcl6^{fl/fl}$ mice, BCL6 insufficiency or derepression of its antagonized targets, such as BLIMP in B cells, likely does not affect tumor immunity (Fig 4D and E). Collectively, these data demonstrate that the BCL6 inhibitor Fx1 controlled tumor progression in a T cell–dependent manner. As lack of BCL6 in activated CD8 T cells stimulated the GLUT3-mediated glycolysis pathway, ultimately retarding CD8⁺ T-cell exhaustion and enhancing their effector functions, we surmise that this mechanism explains the T-cell dependence of Fx1 drug-induced tumor resistance.

## Discussion

T-cell responses are self-limiting, which may be adaptive late in infection when microbial clearance allows cells to recover from exhaustion. However, in the context of antigen persistence, such as in cancer, prevention of exhaustion is often desirable. Here, we show that genetic or pharmacological suppression of BCL6 in activated CD8 cells promotes tumor control. This biological effect is accompanied by enhanced T-cell secretion of functionally

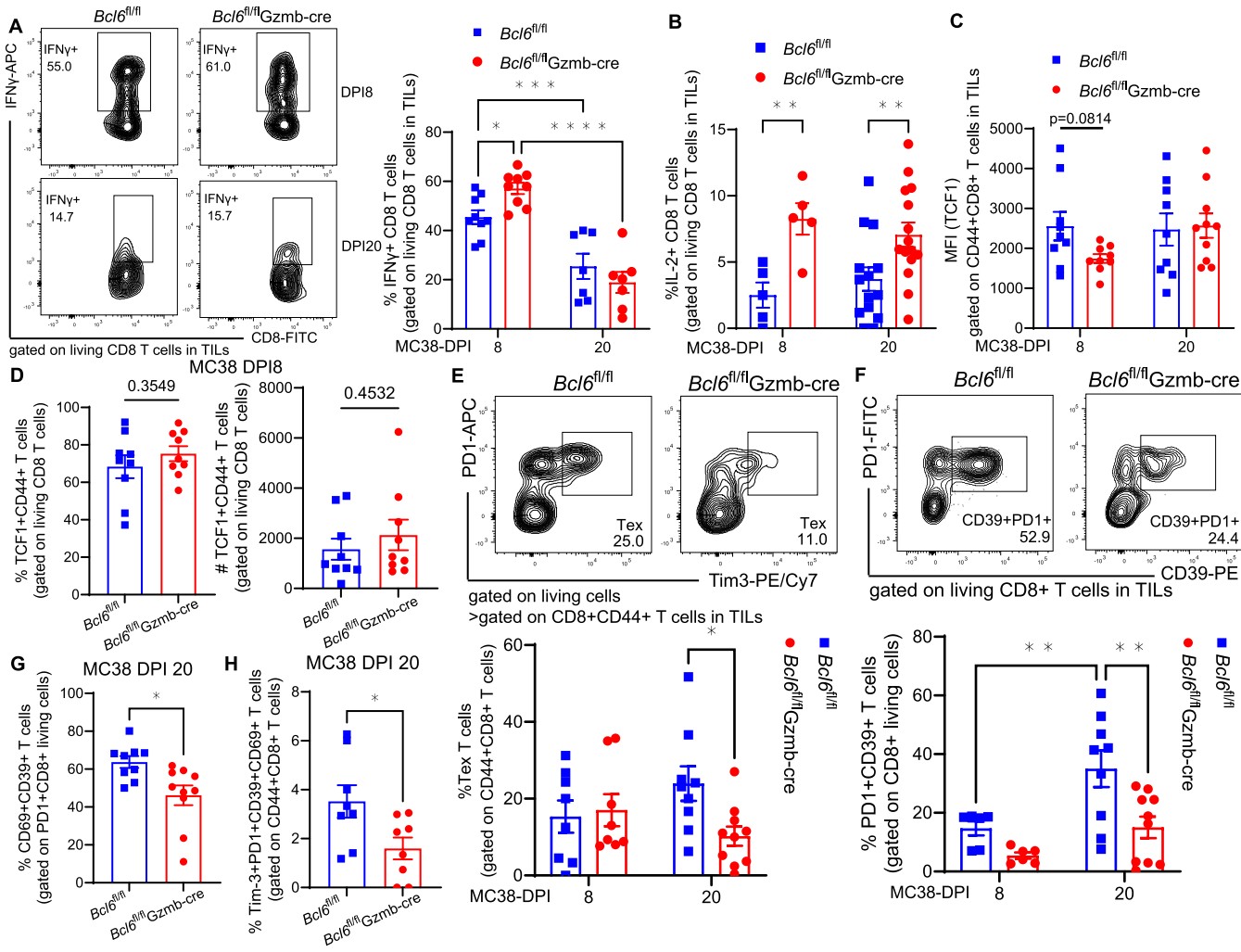

**Figure 2. T-cell dysfunction is lessened in BCL6-deficient CD8 T cells.**
**(A)** Representative flow cytometry plots and frequencies of MC38 tumor-infiltrating IFNγ+ CD8 T cells on DPI 8 (upper and left panel) and DPI 20 (lower and right panel). **(B)** Frequency of IL-2+ CD8 T cells within tumors on DPI 8 (left) and DPI 20 (right). **(C)** Levels of TCF1 expression on tumor-infiltrating CD44⁺CD8⁺ T cells on DPI 8 (left) and DPI 20 (right). **(D)** Proportion (left) and number (right) of tumor-infiltrating TCF1+CD44⁺ CD8 T cells on DPI 8. **(E, F)** Representative flow cytometry plots in the upper panel exhibited the following categories of CD8 T cells on DPI 20: (E) tumor-infiltrating exhausted, PD1+Tim-3+; (F) dysfunctional, PD1+CD39⁺. Lower panels show frequencies of these subpopulations on DPI 8 (left) and DPI 20 (right). **(G, H)** Frequency of tumor-infiltrating PD1+CD69⁺CD39⁺ T cells (G) and PD1+Tim-3+CD39⁺CD69⁺ T cells (H) on DPI 20. Flow cytometry data were pooled from at least three independent experiments. **(A, B, C, D, E, F, G)** Each point in graphs (A, B, C, D, E, F, G) represents the value obtained in an independent mouse. Bars are presented as the mean ± SEM. **(A, B, C, D, E, F, G, H)** P-values were calculated with the two-tailed unpaired t test (B, D, G, H), two-way ANOVA (A, C, E, F). *, **, ***, and **** denote P < 0.05, 0.01, 0.001, and 0.0001, respectively.

important cytokines, such as IFNγ, IL-2, and a reduction in frequency of cells with markers of T-cell exhaustion. The mechanism by which BCL6 normally limits the response appears to be in part through restricting glucose transport and glycolysis in CD8 T cells. Our study supports observations from the study of Chen Dong and colleagues that BCL6 limits T-cell function against cancer (23). However, our mechanistic data provide a distinct interpretation. We show that BCL6 works by directly suppressing a glycolysis stimulation pathway to remodel the effector T-cell function late after the activation of CD8 T cell. We also show that introducing a powerful small molecular drug treatment to suppress BCL6 works even after the tumor grows to nearly 200 mm³ volume. Furthermore, depletion of BCL6 during activation of CD8 T cells did not

result in significant differences in TCF1 expression on CD8 T cells, suggesting that disruption of BCL6 during this time window may not induce stem-like/progenitor-exhausted T cells, thereby inhibiting the generation of terminally exhausted T cells by bypassing the precursor formation pathway. Our analysis is limited to the 3-wk tumor-bearing period, preventing an assessment of longer term effects of BCL6 deficiency or pharmacological repression. BCL6 and TCF1 expression fluctuations are not always correlated, particularly as naïve CD8 T cells highly express TCF1 while barely expressing BCL6, and they are coexpressed ultimately in TOX+ progenitor-like T cells (23). However, lack of BCL6 during CD8 T-cell development and maturation may affect TCF1 base levels in naïve CD8⁺ T cells. Reducing BCL6 levels after

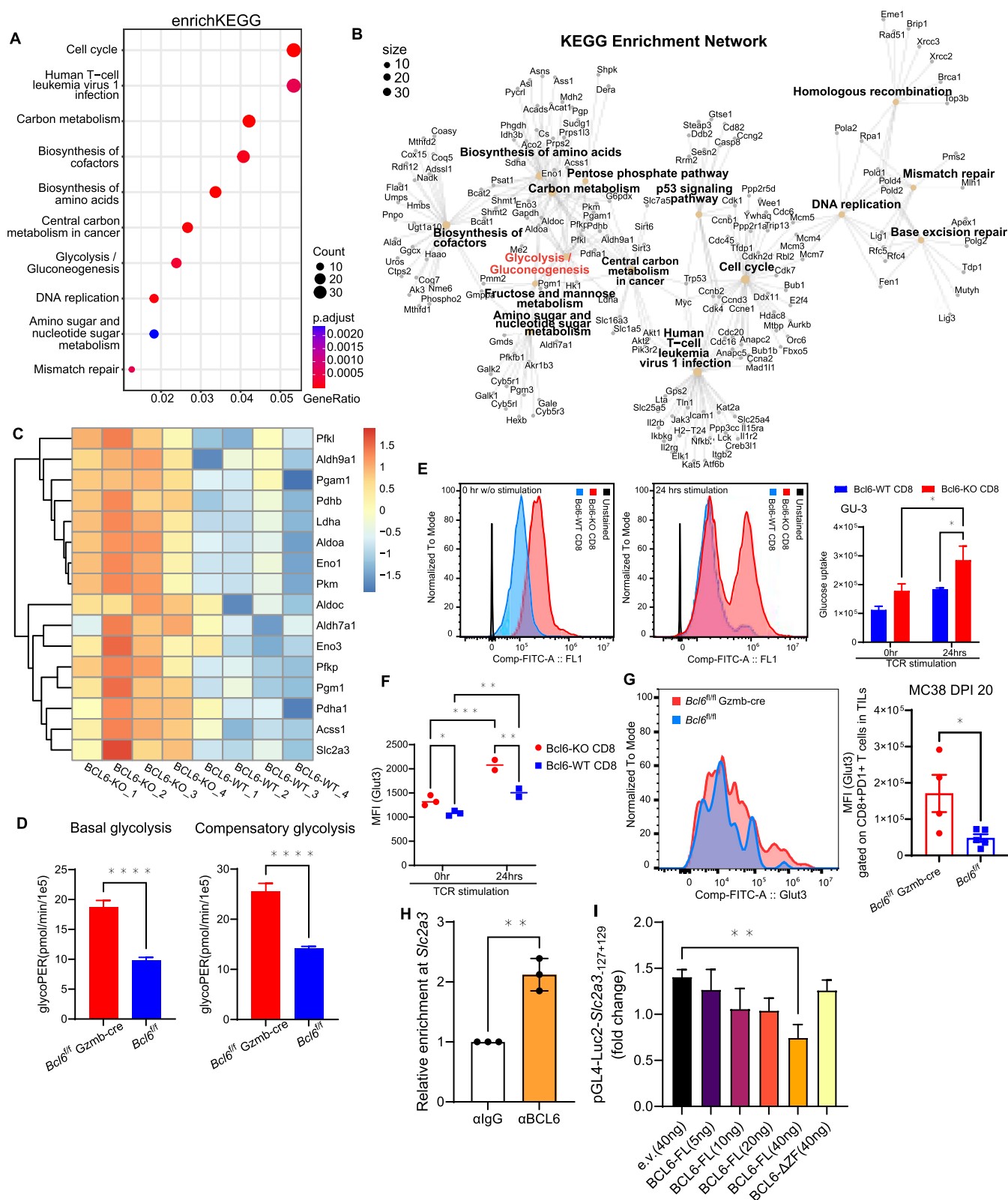

**Figure 3. GLUT3 is repressed by BCL6, in turn regulating CD8 T-cell glycolysis.**
**(A, B, C)** RNA analysis of CD8 T cells isolated from tumors of MC38-inoculated *Bcl6*^fl/fl Gzmb-cre or *Bcl6*^fl/fl mice. **(A, B)** Dot plots and (B) enrichment network of the enrichKEGG analysis for up-regulated genes in activated BCL6-knockout (BCL6-KO) CD8 T cells. **(C)** Heatmap shows the frequencies of remarkable genes in the glycolytic pathway of individual mice from the activated BCL6-KO and BCL6-WT CD8 T-cell groups. **(D, E, F)** Metabolic analysis of CD8 T cells. **(D)** Analysis of basal (left) and compensatory (right) glycolysis levels of BCL6-KO versus BCL6-WT CD8⁺ T cells isolated from *Bcl6*^fl/fl Gzmb-cre or *Bcl6*^fl/fl mice that had been previously activated on anti-

CD8 T-cell activation may promote terminal effector differentiation without affecting the cell subpopulation regulated by TCF1. On the other hand, the model that we used, *Bcl6*[fl/fl] Gzmb-cre, interrupted the powerful function of BCL6 in a narrow cohort of cells in a limited developmental window, leaving other CD8 T cells untouched, and thereby limiting the possibility of affecting cytokine production.

Understanding the mechanisms of T-cell exhaustion should facilitate the identification of pharmacological remedies able to restore the cytotoxicity of dysfunctional T cells to control tumor progression. T-cell exhaustion is complex, difficult-to-manipulate, and still not fully understood. Exhausted T cells cannot be simply identified by their expression of inhibitory receptors, however. Many T-cell effectors secrete cytolytic cytokines while maintaining high levels of inhibitory molecules (28). For example, Tc17 cells express transcription factors STAT3 and RORγt enabling their effector functions by producing IL-17, IL-22, TNF, etc. (28, 29, 30). At the same time, Tc17 cells are increased in Crohn's disease and highlighted by expression of CD6[high], CD39, CD69, PD-1, and CD27[low], which can be targeted to inhibit disease (31). Metabolomics and mapping of cytokine secretions of different CD8 T-cell subtypes may improve our understanding of the development and mechanisms of T-cell exhaustion.

Metabolic reprogramming is essential for the activation and function of effector T cells (14, 32, 33). Upon TCR-mediated activation, T cells increase the expression of glucose transporters and glycolysis enzymes to promote aerobic glycolysis by rapidly stimulating the PI3K/AKT/mTOR1 pathway (34). However, during continuous antigen stimulation exposure to activated T cells, metabolic alterations of glucose, amino acids, and fatty acids are significant features as cells progress from effector T cells to exhausted T cells (32, 34, 35, 36, 37). Shifting from glycolysis to fatty acid oxidation (FAO) in T cells occurs even before T-cell dysfunction occurs, suggesting that the metabolic defect is a trigger rather than a consequence of T-cell dysfunction or exhaustion (17, 38, 39). In this study, we found that inhibition of BCL6 in activated CD8 T cells increased the expression of glucose transporter GLUT3, thereby maintaining glycolysis, and indicating that retarding T-cell exhaustion in tumor conditions is a plausible therapeutic strategy. Administering a BCL6 inhibitor during the optimal time window—before the formation of terminal exhaustion—may help sustain glycolysis in CD8 T cells and serve as an independent, nonmutually exclusive treatment. Currently, the most effective immunotherapy against tumors is immune checkpoint blockade (ICB). A subset of CD8 T cells expressing TCF1, which possesses self-renewal capacity and the potential to differentiate into exhausted cells, has been shown to undergo proliferative burst after PD-1 blockade (8), suggesting that this population gives rise to secondary effector T cells capable of mediating tumor control. However, few studies have investigated the therapeutic efficacy of combining ICB with BCL6 inhibitors; it may be worth further exploration, particularly for treating NSCLC and AML cancer models.

Glucose transporter family members (GLUT) are known to have distinct sequences, substrate affinity, and kinetic properties that reflect specific roles of glucose homeostasis in different cell types (40). In CD4 T cells, GLUT1 is induced upon stimulation and is maintained most predominantly, whereas GLUT3 levels become less prominent, and although GLUT6 also increases, its levels are lower than GLUT1 (41). GLUT1 was reported to be the primary high-affinity glucose transporter of T cells (42). However, compared with activated CD4 effectors, CD8[+] effector T cells are less dependent on GLUT1 and oxygen levels (43). It is reported that the normal function of CD8 effector T cells may use GLUT3 or GLUT6 expression, because the deletion of GLUT1 on CD8 T cells does not result in their inability to differentiate into effector functions and release cytokines such as GzmB, TNFα, IL-2, and IFNγ (41). GLUT3 has a higher affinity and increased transport ability for glucose compared with GLUT1 (44). The overexpression of GLUT3 in murine CD8 T cells promotes glucose uptake and enhances glycogen and fatty acid storage, sustaining effector function (45). Forced expression of GLUT3 along with tumor-specific chimeric antigen or T-cell receptors in engineered human T cells increases their antitumor ability in the context of adoptive T-cell therapies (46). Compared with GLUT1, GLUT3 is induced primarily in the IL-2 signaling branch (47), which also supports our observations of higher levels of IL-2 on BCL6-deleted CD8 T cells. On the other hand, GLUT2, which has been reported to have low affinity to glucose, can be induced during T-cell activation, peaking earlier than GLUT1, and is crucial for maintaining nutrient uptake and metabolism in activated CD8 T cells rather than naïve T cells or activated CD4 T cells (48). Therefore, the functions of GLUT family members on CD8 cells may deserve more experimental digging to sort out their potential roles.

Increased glucose uptake and aerobic glycolysis are primary features of cancer cells as well (49). Aerobic glycolysis is a process in which glucose is converted into pyruvate, which eventually becomes lactate, with a small amount of ATP produced (50). Excessive division, invasion, and metastases of malignant tumor cells alter their energy demands from aerobic respiration to glycolysis, resulting in enhanced production of glycolytic enzymes and glucose transporter proteins (51). Oncogenic transformation in cultured tumor cells causes strong glucose transport and increased expression of GLUT1 and GLUT3 (52). Positive correlations between glucose uptake and levels of GLUT1, GLUT3, or GLUT12 were observed in many different types of cancers that are associated with poor survival rates (53). These findings indicate that any antitumor

CD3/anti-CD28 plates overnight, then stimulated for 48 h with IL-2 and anti-CD28. **(E, F)** Comparison of glucose uptake and GLUT3 expression in *Bcl6*[fl/fl] CD4-cre CD8+T cells (BCL6-KO) versus *Bcl6*[fl/fl] CD8+T cells (BCL6-WT). **(E)** Elevated glucose uptake in BCL6-KO compared with BCL6-WT CD8 T cells after 24 hrs of TCR stimulation. Similar differences were reproduced in a second experiment. **(F)** Levels of GLUT3 expression on BCL6-KO and BCL6-WT CD8 T cells before and after TCR stimulation. Data were pooled from two to three independent experiments. **(G)** Increased GLUT3 detected on tumor-infiltrating PD1+ CD8 T cells in *Bcl6*[fl/fl] Gzmb-cre mice on DPI 20. Each point represents the value obtained in an independent mouse. **(H)** Analysis of BCL6 binding to the promoter regions of *Slc2a3*, which encodes GLUT3 in mice. Binding activity was confirmed three times independently using CD8 T cells isolated from spleen of WT mice. **(I)** Luciferase assay showing that full-length (FL) BCL6 repressed the expression of the reporter gene (luc2) driven by the *Slc2a3* promoter, and that the inhibition of luc2 signal was lost when BCL6 lacked its DNA binding activity (BCL6-ΔZF). Data were pooled from four independent experiments. Bars show the mean ± SEM. **(D, E, F, G, H, I)** *P*-values were calculated with the two-tailed unpaired *t* test (D, G, H), two-way ANOVA (E, F, I). *, **, ***, and **** denote *P* < 0.05, 0.01, 0.001, and 0.0001, respectively.

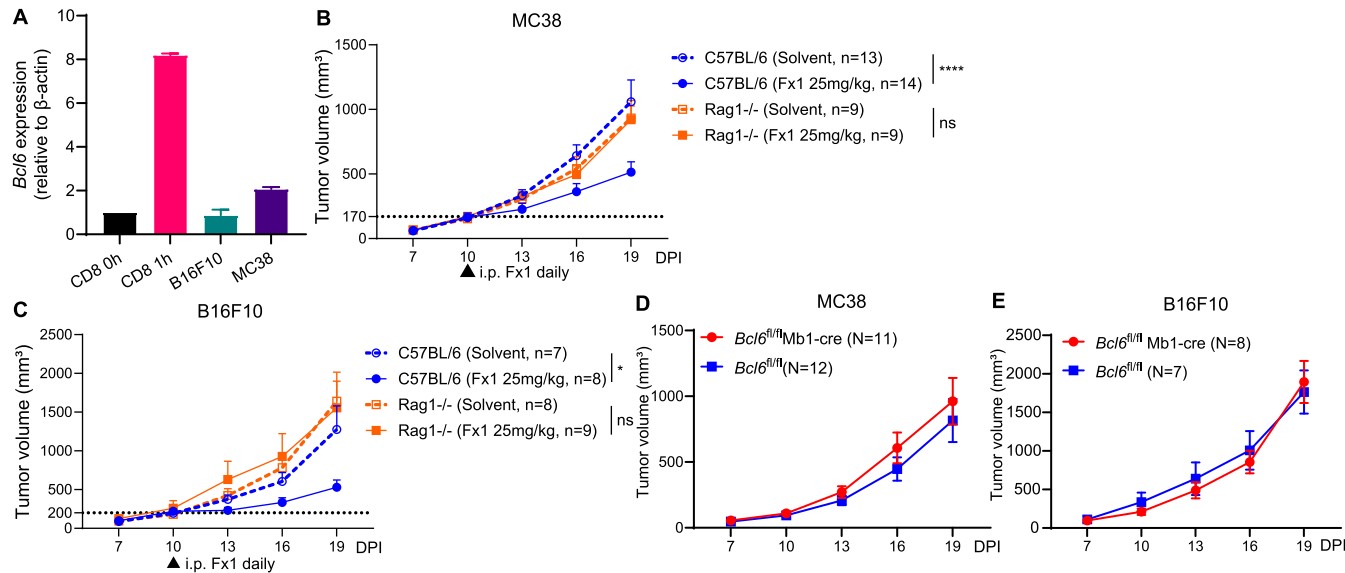

**Figure 4. Antitumor efficacy of the BCL6 inhibitor Fx1 depends on T cells.**
**(A)** Relative expression of BCL6 in B16F10 and MC38 tumor cells is comparable to the level in CD8 T cells without TCR stimulation, and BCL6 is quickly increased in CD8 T cells after activation. **(B, C)** Analysis of tumor growth in C57BL/6 and $Rag1^{-/-}$ mice implanted with MC38 (B) or B16F10 (C) tumor cells and receiving Fx1 treatment i.p. daily starting from DPI 10. **(D, E)** Analysis of tumor growth of MC38 (D) and B16F10 (E) in $Bcl6^{fl/fl}$ Mb1-cre and control $Bcl6^{fl/fl}$ mice. Bars show the mean ± SEM. **(B, C)** P-values were calculated with the two-way ANOVA (B, C). * and **** denote $P < 0.05$ and $0.0001$, respectively, and ns denotes $P > 0.05$.

therapy targeting glycolysis in the tumor microenvironment is a double-edged sword requiring enhancement of the glycolytic pathway to favor effector CD8 T cells while excluding side effects that promote the aggressive progression of tumor cells. BCL6 directly represses genes involved in the glycolysis pathway, including *Slc2a1* and *Slc2a3* (54). Although BCL6 associates with the promoters of many genes of this family (55), there is little functional information available about other members. Increasing studies suggest that BCL6 works as an oncogene in human cancer (56). BCL6 is a powerful transcriptional repressor that silences a variety of genes involved in the DNA damage response, cell cycle, and death, such as TP53, CDKN1A/B, ATR, CHEK1, CDKN2A/B, PTEN (57). In the present study, the antitumor efficacy of Fx1 relied on immune cells, suggesting BCL6 expression in MC38 or B16F10 tumor cells likely did not reflect a direct antitumor effect, though inhibiting BCL6 in tumors may benefit tumor growth resistance as well. Ultimately, our data introduce an effective therapeutic strategy by targeting effector CD8 T cells to improve their glycolysis, thereby prolonging their antitumor cytotoxic functions.

# Materials and Methods

## Mice and tumor models

$Bcl6^{fl/fl}$ conditional knockout mice ($Bcl6^{fl/fl}$ CD4-cre, $Bcl6^{fl/fl}$ Mb1-cre, $Bcl6^{fl/fl}$ Ox40-cre, and $Bcl6^{fl/fl}$ Gzmb-cre) were generated and maintained in-house. WT C57BL/6 and $Rag1^{-/-}$ ($Rag1^{tm1}$Mom) mice were obtained from Jackson Labs. Mice were used at 6–8 wk of age, and all procedures adhered to institutional animal care guidelines.

Subcutaneous tumor models were established by inoculating $1 \times 10^6$ MC38, LLC1, or B16F10 cells per mouse. Tumor growth was monitored every 3 d using digital calipers.

## Reagents and antibodies

All cell culture experiments used DMEM (Thermo Fisher Scientific) supplemented with 10% FBS (Gibco). Fluorescently conjugated antibodies for flow cytometry were purchased from BD Biosciences. CUT&RUN Assay Kit was obtained from Vazyme, and RNA-sequencing libraries were prepared using SMART-Seq v4 Ultra Low Input RNA Kit (Takara Bio). For qPCR, reverse transcription was performed using HiScript III RT SuperMix (Vazyme), and real-time PCR used qPCR SYBR Green Master Mix (Vazyme). Fx1 (Selleckchem) was dissolved in a vehicle containing 40% DMSO, 40% PEG3000, and 20% PBS, and administered intraperitoneally at 20 mg/kg.

## Flow cytometry analysis

Tumor-infiltrating lymphocytes (TILs) were isolated by enzymatic digestion of tumor tissue with collagenase I (1 mg/ml) and DNase I (0.1 mg/ml). Cells were stained in FACS buffer (PBS supplemented with 2% FBS) and analyzed on a BD LSRFortessa flow cytometer. Data were processed using FlowJo software, and gates were set based on fluorescence-minus-one (FMO) controls.

## RNA sequencing

CD8+PD1+ TILs were sorted to >95% purity. RNA was extracted, and libraries were generated using SMART-Seq v4 Kit. Sequencing was conducted on an Illumina NovaSeq platform, with differential gene

expression analyzed using DESeq2. Enrichment analyses were performed with the KEGG pathway database.

## Metabolic assays

Extracellular acidification rates (ECARs) were measured using Seahorse XF Glycolysis Stress Test Kit (Agilent) on XF96 Analyzer. For glucose uptake assays, cells were cultured in glucose-free medium and incubated after CD3/CD28 stimulation.

## Chromatin immunoprecipitation (CUT&RUN)

CD8 T cells were isolated from the spleens of WT mice using EasySep Mouse CD8$^+$ T Cell Isolation Kit (StemCell Technologies), and their lysates were used for ChIP assay directly. CUT&RUN assays were performed as described by the manufacturer of Hyperactive pG-MNase CUT&RUN Assay Kit (Vazyme). Anti-BCL6 (Cell Signaling Technology) and isotype control antibodies (Cell Signaling Technology) enriched DNA–protein complexes. qPCR was conducted to assess binding at the *Slc2a3* promoter region, with enrichment quantified relative to IgG controls. Primers for qPCR were as follows: Forward GGAGCGGTGAAGATCAGATAAG; Reverse GAAGCCCAGCCTACCTATTT.

## Luciferase reporter assay of *Slc2a3* gene transcription

On the day of transfection, EL4 suspension cells were seeded at a density of $5 \times 10^4$/well in a 96-well plate (clear bottom) first. To a sterile tube, Opti-MEM was added to make the final volume 50 $\mu$l after adding the transfection-grade DNA (10 $\mu$l/well). 200, 100, 50, or 25 ng of pCMV-2b-mBCL6-FL plasmid or an equal amount of the empty vector was added into the medium followed by 250 ng reporter plasmid pGL4.10[Luc2]-Slc2a3(–127,+129) and 50 ng internal control vector pGL4.74[hRluc/TK]. For a 3:1 ViaFect transfection reagent (Promega): DNA ratio, we added 1.5 $\mu$l of the reagent to the media, mixed immediately, then waited for 10 min at RT. To each well, 10 $\mu$l of the mixture was added to the plate, with gentle pipetting, and then, the plate was returned to the incubator. After 48 h, we followed the instructions of the Dual-Glo Luciferase assay system (Promega) to capture the firefly and Renilla luminescence in a luminometer and calculate the fold changes.

## Statistical analysis

Data are presented as the mean ± SEM. Statistical significance was determined using GraphPad Prism 9.0, employing unpaired two-tailed *t* tests or two-way ANOVA with multiple comparison corrections where applicable. Statistical thresholds for significance were set at *P* < 0.05.

# Data Availability

All data are summarized in the figure and table files of this study. Further information can be obtained upon request from the authors.

# Supplementary Information

# Acknowledgements

This study was supported by the National Institutes of Health (R01AI137252 to C Xiao and D Nemazee). We thank members of Xiao and Nemazee laboratories for their discussion.

## Author Contributions

F Luan: conceptualization, data curation, formal analysis, and investigation.
Y Li: conceptualization, data curation, formal analysis, supervision, validation, investigation, methodology, and writing—original draft, review, and editing.
J Ning: methodology.
JT Tran: resources and investigation.
TR Blane: resources and investigation.
R Bhargava: resources and investigation.
Z Huang: conceptualization.
C Xiao: conceptualization, funding acquisition, and project administration.
D Nemazee: conceptualization, supervision, funding acquisition, project administration, and writing—original draft, review, and editing.

## Conflict of Interest Statement

The authors declare that they have no conflict of interest.

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
