## [Reviewer comments · Life Science Alliance]

Loss of Bcl6 promotes anti-tumor immunity by activating glycolysis to rescue CD8 T cell function

Fangkun Luan, Yunqiao Li, Jia Ning, Jenny Tran, Tanya Blane, Raag Bhargava, Zhe Huang, Changchun Xiao, and David Nemazee

DOI: <https://doi.org/10.26508/lsa.202503335>

Corresponding author(s): David Nemazee, Scripps Research Institute and Yunqiao Li, Scripps Research Institute

Review Timeline:

Submission Date:	2025-03-31
Editorial Decision:	2025-05-08
Revision Received:	2025-08-15
Editorial Decision:	2025-09-15
Revision Received:	2025-10-14
Accepted:	2025-10-15

Scientific Editor: Tim Fessenden

Transaction Report:

May 8, 2025

Re: Life Science Alliance manuscript #LSA-2025-03335

Prof. David Nemazee
Scripps Research Institute
Department of Immunology
10550 Torrey Pines Road, IMM-29
La Jolla, CA 92037

Dear Dr. Nemazee,

Thank you for submitting your manuscript entitled "Lack of Bcl6 promotes anti-tumor immunity by activating glycolysis to rescue CD8 T cell function" to Life Science Alliance. The manuscript was assessed by expert reviewers, whose comments are appended to this letter. We invite you to submit a revised manuscript addressing the Reviewer comments.

As you will see, reviewers uniformly appreciated these observations that relate the activity of BCL6 with metabolism and tumor immunity through effects on CD8 T cells. Reviewers made several important suggestions to better support the main conclusions as well as properly contextualize these findings given previous work on BCL6. Namely, Reviewer 1 remarked that the work leaves it unclear if BCL6 acts solely through repression of GLUT3 in this setting. A revised manuscript should address this either by knocking out GLUT3 or by tempering conclusions accordingly. This reviewer also raised the possibility that BCL6 acts during T cell priming, a point echoed by Reviewer 3. A revised manuscript should test this using the inhibitor Fx1. Reviewers 2 and 3 noted that conclusions on exhausted/dysfunctional T cells should be supported with improved flow cytometry staining. These reviewers both requested that claims of BCL6 expression in intratumoral CD8 T cells should be supported with tissue staining, both to confirm loss of this protein in CD8 T cells and to detect its presence in CD4 T cells (in control tumors). Related to this point, Reviewer 3 noted that the data shown in Fig 1A do not support a role for BCL6 in T cells, and here we concur these data should be swapped with those from Fig S1. This reviewer also made an intriguing suggestion on TCF1 levels, whose resolution may better align the present observations with prior work that relates BCL6 with memory T cells. Finally Reviewer 3 made several helpful suggestions to improve the text. Additional experimental data beyond those noted here are not required in a revision.

The typical timeframe for revisions is three months. Please note that papers are generally considered through only one revision cycle, so strong support from the referees on the revised version is needed for acceptance. When submitting the revision, please include a letter addressing the reviewers' comments point by point.

We hope that the comments below will prove constructive as your work progresses. Thank you for this interesting contribution to Life Science Alliance. We are looking forward to receiving your revised manuscript.

Sincerely,

- A letter addressing the reviewers' comments point by point.
- An editable version of the final text (.DOC or .DOCX) is needed for copyediting (no PDFs).

B. MANUSCRIPT ORGANIZATION AND FORMATTING:

Reviewer #1 (Comments to the Authors (Required)):

The manuscript provides data that the co-repressor BCL6, when knocked out in CD8+ T cells, promotes glycolysis, decreases exhaustion, and leads to better effector function and tumor control in transplantable tumor models. The studies utilize BCL6-floxed mice and granzyme-cre, CD4-cre, OX49-cre, Mb1-cre, and RAG mice to demonstrate that BCL6 in CD8+ T cells but not in CD4+, NK, or B cells play a role in antitumor immunity. The study also uses a BCL6 inhibitor as a pharmacologic treatment of cancer. Human data confirm that patients with tumors with low BCL6 have higher survival. This conclusively demonstrates that BCL6 restricts CD8+ tumor immunity and inhibition of BCL6 could be a viable therapeutic option. This advances the field of cancer immune therapy and CD8+ T cell differentiation.

The main point that BCL6 restricts CD8+ T cell antitumor activity is strongly supported by the data. Additional experiments to show this are not needed.

The mechanism of BCL6 as a repressor of GLUT3 is shown by luciferase reporter assays, RNAseq studies of BCL6-deficient CD8+ T cells, GLUT3 qrtPCR measurements, and glycolysis assays. These support a role for BCL6 in regulating GLUT3 but do not demonstrate that this activity underlies the increased anti-tumor activity of BCL6-deficient CD8+ T cells. Improved glycolysis is known to increase CD8+ antitumor activity, but is this the sole mechanism of BCL6 activity in CD8+ T cells? An experiment in which GLUT3 is knocked out in BCL6-KO CD8+ T cells might show if regulating GLUT3 is required for BCL6's restraining of CD8+ antitumor activity.

The possibility that BCL6 promotes exhaustion in CD8+ T cells in tumors is supported by findings that BCL6-deficient intratumoral CD8+ T cells demonstrate slightly higher effector activity and lower expression of exhaustion/dysfunction markers (PD1, TIM3, CD39). This could be due to smaller tumor sizes and changed tumor microenvironment. Whether BCL6 regulates transcription factors thought to be involved in exhaustion is not known, but the RNAseq data suggest that it does not. The relationship between glycolysis/GLUT3 and exhaustion is also not demonstrated. It would be good to look at whether overexpression of GLUT3 in CD8+ T cells reverses exhaustion.

The studies with the inhibitor show therapeutic potential and suggests that BCL6 in effector CD8+ T cells is restraining activity. The phenotype seems weaker than the genetic deletion. This might suggest a role for BCL6 during priming. Perhaps giving the inhibitor earlier would test if BCL6 plays a role in priming.

No other issues to address.

Reviewer #2 (Comments to the Authors (Required)):

In this study, Luan et al. demonstrate that BCL6 suppresses the anti-tumor activity of CD8+ T cells. Using a series of T cell specific Cre drivers, the authors show that BCL6 knockout in CD8+ T cells enhances tumor control in a subcutaneous model across multiple cell lines. This improved control appears to be mediated by increased effector function and reduced exhaustion

of CD8⁺ T cells. Notably, the GzmB-Cre model used deletes BCL6 upon activation, which may explain why the authors did not observe differences in the TCF1⁺ population previously observed. Transcriptomic analysis of BCL6-deficient CD8⁺ T cells suggested metabolic alterations, including increased glycolysis, suggesting a mechanistic basis for enhanced cellular fitness. The authors convincingly demonstrated that glycolysis is improved in BCL6 deficient T cells. Finally, the authors demonstrate that pharmacologic inhibition of BCL6 after tumor initiation can improve anti-tumor immunity. Overall, the manuscript is clearly written and presents a well-executed study on the role of BCL6 in the anti-tumor response of CD8⁺ T cells. While the novelty of targeting BCL6 to enhance anti-tumor responses has been somewhat diminished by the Sun et al. publication (Science Immunology 2023), the authors take a distinct approach and provide additional insight identifying altered metabolism as a mechanism contributing to increase fitness in the BCL6 knockout cells. Although their identification of GLUT3 as a target to improve T cell fitness has already been confirmed in CAR-T cell mouse models. Given the differences between the previous model and the total knockout in transgenic T cells previously published, it would be informative to understand the expression of Bcl6 and its loss in the author's system. The authors are to be commended for the rigor of their work, including clear reporting of statistics, mouse numbers, and replicates.

Comments to the authors:

1. As the model used a Cre controlled under the GzmB promoter rather than an E8I-Cre, BCL6 is lost only upon activation and its knockout is not restricted to CD8 T cells. Although it appears that BCL6 is not expressed highly in naïve CD8s it would be good to know if BCL6 is expressed by any CD8s in the tumor microenvironment, or if the promoter essentially functions as a total knockout in the CD8 compartment. Additionally, although GzmB is associated more with CD8s, CD4 T cells can express GzmB when activated. Staining T cells for BCL6 from the tumor microenvironment would also assess the degree to which the CD4 compartment loses BCL6 expression under this promoter.
2. In Figure 4B, the tumors in Rag1 knockout mice appear to have the same growth kinetics as the WT mice. I found this interesting as it has been reported that the adaptive immune system contributes some at baseline to control of MC38 tumors (e.g. Efremova et al. Nat. Com 2018). I am curious if the authors normally see no difference in growth between WT and Rag1 KO mice with their inoculating dose and if they do see a difference when taking the tumor out longer?
3. In Figure 2, I hesitate to define a T cell as exhausted solely based on co-expression of PD-1 and TIM-3, or dysfunction based on PD-1 and CD39 expression. These markers are associated with exhaustion but are not the definition. It could strengthen the interpretation to include an analysis showing the proportion of T cells that are single-, double-, or triple-positive for these inhibitory receptors, providing a better view of the exhaustion spectrum within the population.
4. In Figure 3B, it may just be my copy, but some gene names appear clipped off ensure that all are readable in the final image.

Reviewer #3 (Comments to the Authors (Required)):

In this study, Luan et al. demonstrate a role of BCL6 in the antitumor CD8⁺ T cell response in flank transplant tumor models, and show that BCL6 knockout enhances effector functions and decreases the proportion of exhausted T cells. They also report that BCL6 represses GLUT3 by directly binding to the Slc2a3 locus, and that BCL6 knockout thus increases GLUT3 and enhance glycolysis in these T cells, which may underlie their enhanced function and decreased exhaustion. Finally, the authors use an inhibitor of BCL6, Fx1, to show that this checkpoint can be pharmacologically targeted to enhance antitumor T cell responses in vivo. The study is mostly well-executed and the manuscript clearly written. However, the results are largely derivative in that the role of BCL6 in CD8 T cells has been studied in a number of publications in both viral and cancer models, and a role for BCL6 in repressing glycolysis via repression of Slc2a3 in CD4 helper T cells (Th1) has been previously reported (Oestreich, KJ, et al., Nature Immunology, 15(10):957-964, 2014). Nevertheless, the present manuscript provides an important angle on cancer immunotherapy and would still be valuable to the field, but there are a few critical points that must be addressed prior to publication.

1) The human TCGA analysis correlating BCL6 expression (bulk RNA) to overall survival is problematic and the conclusions drawn (that high BCL6 expression indicates infiltration by dysfunctional T cells) are not supported by these data. This analysis should either be removed or replaced by one that is more nuanced, with IHC/IF staining of T cells, or analysis of single cell datasets that can enable specific enumeration of BCL6-expressing T cells.

2) The authors mention some prior work on the role of BCL6 in T cells, but review of this prior literature is incomplete. The authors should more thoroughly integrate the prior literature into their introduction and discussion, and provide a nuanced discussion of how their findings fit into this broader picture. Specifically, these two papers should be discussed more thoroughly and with comparison to the present manuscript's results: Liu, Z, et al., J Immunol (2019) 203 (2): 323-327, Sun, Q, et al., and Science Immunology (2023) 88 (8). While these studies come to slightly different conclusions regarding the role of BCL6 in generating versus maintaining memory T cells, they agree on the importance of this gene in driving expression of TCF1 and supporting the memory progenitor transcriptional program while antagonizing expression of effector genes.

It would seem that Luan et al., in the present study, have generated results consistent with these prior studies, but have focused only on the enhancement of the effector response early on, while avoiding the question of memory responses, persistence of BCL6^{-/-} T cells, and longer-term immunity. While it may be outside the scope of the present study to perform memory response/recall experiments, the authors should discuss these nuances in detail. For example, while BCL6 knockout may enhance the early effector response by promoting terminal effector differentiation, this may be at the expense of the longer-term memory response, something that is not accurately recapitulated in short-term cell line transplant models.

Specific Comments

1)

54-55 "Compared to effector or memory T cells, the category of exhausted T cells encompasses a broad definition,"

A little misleading, memory and effector T cells also come in many types/definitions. I suggest eliminating the "compared to effector or memory T cells" clause.

2)

75 "BCL6 is also critical for maintaining and generating memory CD8 T cells [21]"

The results of this study [21] are somewhat contradictory to the messaging of the present manuscript-that BCL6 deletion can INCREASE the quality and function of the antitumor T cell response. If generation and maintenance of memory CD8 T cells is impaired by BCL6 knockout, then this would seem a poor strategy to increase antitumor immunity. However, this cited paper has the major caveat of using germline BCL6 knockout mice, which could affect hematopoiesis and all stages of T cell (and other immune cell) development, potentially resulting in phenotypically/functionally different naïve CD8 T cells used in the transfer studies. In addition, there is another paper that reported that conditional BCL6 knockout in CD8 T cells is required for the generation, but not maintenance, of memory CD8 T cells (*J Immunol* (2019) 203 (2): 323-327), although another study reported that BCL6 deficiency impairs the persistence, but not generation, of memory progenitor CD8 T cells (Sun, Q, et al., and *Science Immunology* (2023) 88 (8)). Despite these somewhat conflicting results, both of these studies align in their report that BCL6 positively regulates the expression of the Tcf7 gene to support the memory progenitor state.

The authors should discuss these prior findings in much more detail and how their work fits into this broader literature:

Specifically, the authors report that *Bcl6*-flox/flox; *Gzmb*-Cre CD8 T cells in the MC38 model do not show decreased TCF1 expression, which would conflict with the results of the prior studies mentioned above. However, looking closely at Figure 2C, it appears that there actually is a (probably) significant decrease in TCF1 expression in these CD8 T cells at day 8, but not 20. The authors should report statistical significance here, and even if not significant, mention the trend and rationalize their results with the prior literature. MFI is also a rather crude way of plotting these data. The authors should also show percent or total CD8⁺/CD44⁺ TILs that are TCF1⁺. Is it possible that BCL6 knockout is skewing the balance of memory progenitor to effector ratio towards more short-lived effector cells, and thus early on in the response there are more TCF1 low effector T cells? Whereas later in the response, the short-lived effector T cells have contracted (consistent with the Sun, Q, et al. study above reporting that these *Bcl6*^{-/-} effectors have impaired persistence), leaving only the memory/memory progenitor cells with high levels of TCF1? It would also be helpful to assess the degree/efficiency of BCL6 knockout at the 8-day and 20-day timepoints in the model, as there may be selection against BCL6 knockout later in the response, given reported defects in persistence, resulting in enrichment of BCL6^{+/+} or BCL6^{+/-} CD8 T cells that failed to undergo complete recombination. As mentioned in another comment, it is important to validate that the Cre-recombinase system is actually resulting in efficient deletion of BCL6 in CD8 T cells at the timepoints analyzed.

3)

75-78 "The expression level of BCL6 exhibits a strong positive correlation with the number of central memory CD8 T cells (TCM) upon antigen stimulation and contributes to antigen-specific TCM secondary expansion"

Expression level in what context? This sentence confusing. Are the authors saying that the expression level of BCL6 in central memory T cells is strongly correlated to other markers associated with central memory? Or the expression level of BCL6 in T cells prior to activation correlates with their subsequent development into central memory? Or that the expression level of BCL6 in T cells following antigen stimulation correlates with their subsequent development into central memory T cells? Seems like probably the last, but the wording could be clearer.

4)

156-163 "We found that tumor cases with lower levels of *Bcl6* mRNA had favorable overall survival rates in all types of cancer (Fig. 1A) and this phenotype was confirmed in colon adenocarcinoma (COAD, with $p=0.049$), pancreatic adenocarcinoma (PAAD, with $p=0.021$) and, possibly, rectum adenocarcinoma (READ, no significant difference but trend was shown) (Fig. S1A-161 C). Moreover, tumor-infiltrating CD8 T cell levels and *Bcl6* expression showed a positive correlation in COAD, PAAD and READ samples (Fig. S1D-F), suggesting that BCL6 promotes CD8 T cell proliferation within tumors while suppressing their killing function."

This analysis is fraught and the conclusions counterintuitive. There are numerous publications demonstrating that levels of CD8

T cell infiltration within solid tumors like the ones analyzed here are associated with better survival. It is also impossible in this analysis to determine if BCL6 expression is coming from CD8 T cells or tumor/stromal/other immune cells. To support the authors' hypothesis convincingly, it would be important to show that BCL6 level in tumors is driven mostly by CD8 T cells, and/or that BCL6+ CD8 T cells (via IF/IHC, flow, or scRNA-seq) correlates with survival or some other feature strongly associated with survival. Otherwise, the argument is very weak. One possible explanation is that tumor intrinsic expression of BCL6 is associated with a more malignant phenotype, despite these tumors having a slightly elevated level of T cell infiltration (relatively weak Rho), or these T cells are excluded at the edges of the tumor.

5)

190-191 "Moreover, the CD39+PD1+ and CD69+PD1+ dysfunctional CD8 T cells"

Why are CD69+PD1+ CD8s being called dysfunctional? This could also just indicate a resident effector phenotype or a recently activated phenotype.

6)

193-197 "These data suggest that as the size of inoculated tumors increase the function of CD8 T cells transforms from effector to exhaustion or dysfunction, and that the elimination of BCL6 in activated CD8 T cells can reverse this process to some extent, allowing them to exert their killing effector for a longer time."

This is a strange way to interpret the data. Do the authors actually show that the size of the tumor is associated with exhaustion phenotypes? And is that really the point? A better way to say this would be that in the face of an actively growing tumor and persistent antigen stimulation, the infiltrating T cells become progressively exhausted/dysfunctional. But this is a well-appreciated concept, and doesn't really need to be said. It suffices to say that BCL6 deletion reduces the acquisition of features of exhaustion in tumor-infiltrating CD8 T cells in the model. The authors should also be precise here with wording related to reversal versus prevention of exhaustion. Given the early time points assessed in the model, the experimental results support a role of BCL6 knockout in preventing exhaustion at the time point analyzed, not reversing established exhaustion.

7)

235 "We first confirmed that BCL6 expression was strongly induced upon T cell activation (Fig. 4A)."

What about BCL6 levels in T cells treated with Fx1? Is this inhibitor actually specific and on-target? It should be very straightforward to assess the effects of Fx1 treatment on BCL6 expression levels in T cells in vitro, and would also be nice to see in the in vivo experiments at the doses used in the tumor growth studies.

8)

303-304 "Targeting exhausted T cells to restore their effector function is an independent, non-mutually exclusive approach clearly worth further investigations."

Immune checkpoint blockade also targets exhausted T cells, albeit through a different mechanism, so the first clause of this sentence is not accurate. Are the authors drawing the distinction that ICB targets progenitor exhausted T cells while BCL6 inhibition/Fx1 is specifically targeting terminally exhausted T cells? The distinction is not clear, and the authors have not clearly shown that BCL6 inhibition is acting in any way on terminally exhausted T cells to rescue their function. Rather, BCL6 inhibition may be preventing the onset of exhaustion specifically in the proliferating effector population. Alternatively, it is possible that BCL6 knockout is resulting in the early contraction/apoptosis of exhausted T cells, based on the impaired persistence of BCL6 knockout effector CD8 T cells reported in Sun, Q, et al., and therefore there could be fewer exhausted T cells left (another reason that looking at overall numbers/magnitude of the T cell response and these different populations between experimental groups is really important). Importantly, the authors start treatment with Fx1 at day 10, which is likely close to the peak of priming and proliferative expansion of effector T cells in the model, and not a time point when many T cells would already be exhausted, making it impossible to disentangle effects on exhausted T cell generation versus existing exhausted T cells.

This raises an important point that the authors should address in their Discussion. Does BCL6 inhibition prevent exhaustion or rescue it, and if only the former, it is unlikely to work effectively in established tumors due to the majority of infiltrating T cells already being dysfunctional. This also raises a major limitation of flank transplant models-the kinetics are too rapid to easily separate treatment effects across distinct phases of the T cell response. It would also be worth entertaining the possibility that BCL6 inhibition could synergize with ICB, if the above mechanism proves true, as ICB would enable proliferative expansion of early-exhausted T cells with effector functions from progenitor exhausted T cells, and BCL6 inhibition may prolong their

proliferative and effector function through maintaining glycolysis and proliferation. To be clear, these nuances should be entertained in the Discussion and no additional experiments are being requested to resolve these specific questions.

9)

Given the well-established role of BCL6 in the cell cycle and the fact that rapidly proliferating cells are associated with glycolytic metabolism, the authors should assess the effects of BCL6 knockout/inhibition on T cell proliferation using EDU or other proliferation assays, and by quantifying/reporting the magnitude of the T cell response in the in vivo models. Is proliferation enhanced with BCL6 knockout, and does this result in greater numbers of primed T cells in LNs/spleen/or blood and increased tumor-infiltrating CD8 T cells?

10)

The authors should show that their BCL6 knockout CD8 T cells actually have loss of BCL6 protein, by Western, flow, or other methods. It would be important to include the controls without Cre lines, and also CD4 T cells. This is quite important to establishing that the model is actually functioning as intended and the phenotypes are indeed due to BCL6 knockout, especially in the later timepoints where there may be selection against BCL6 knockout CD8 T cells due to impaired persistence.

Re: Life Science Alliance manuscript #LSA-2025-03335

Prof. David Nemazee

Scripps Research Institute

Department of Immunology

10550 Torrey Pines Road, IMM-29

La Jolla, CA 92037

Dear Dr. Nemazee,

Thank you for submitting your manuscript entitled "Lack of Bcl6 promotes anti-tumor immunity by activating glycolysis to rescue CD8 T cell function" to Life Science Alliance. The manuscript was assessed by expert reviewers, whose comments are appended to this letter. We invite you to submit a revised manuscript addressing the Reviewer comments.

As you will see, reviewers uniformly appreciated these observations that relate the activity of BCL6 with metabolism and tumor immunity through effects on CD8 T cells. Reviewers made several important suggestions to better support the main conclusions as well as properly contextualize these findings given previous work on BCL6. Namely, Reviewer 1 remarked that the work leaves it unclear if BCL6 acts

solely through repression of GLUT3 in this setting. A revised manuscript should address this either by knocking out GLUT3 or by tempering conclusions accordingly. This reviewer also raised the possibility that BCL6 acts during T cell priming, a point echoed by Reviewer 3. A revised manuscript should test this using the inhibitor Fx1. Reviewers 2 and 3 noted that conclusions on exhausted/dysfunctional T cells should be supported with improved flow cytometry staining. These reviewers both requested that claims of BCL6 expression in intratumoral CD8 T cells should be supported with tissue staining, both to confirm loss of this protein in CD8 T cells and to detect its presence in CD4 T cells (in control tumors). Related to this point, Reviewer 3 noted that the data shown in Fig 1A do not support a role for BCL6 in T cells, and here we concur these data should be swapped with those from Fig S1. This reviewer also made an intriguing suggestion on TCF1 levels, whose resolution may better align the present observations with prior work that relates BCL6 with memory T cells. Finally Reviewer 3 made several helpful suggestions to improve the text. **Additional experimental data beyond those noted here are not required in a revision.**

The typical timeframe for revisions is three months. Please note that papers are generally considered through only one revision cycle, so strong support from the referees on the revised version is needed for acceptance. When submitting the revision, please include a letter addressing the reviewers' comments point by point.

Thank you for this interesting contribution to Life Science Alliance. We are looking forward to receiving your revised manuscript.

Sincerely,

Tim Fessenden

Scientific Editor

Life Science Alliance

B. MANUSCRIPT ORGANIZATION AND FORMATTING:

Full guidelines are available on our Instructions for Authors page,

<https://www.life-science-alliance.org/authors>

Reviewer #1 (Comments to the Authors (Required)):

The manuscript provides data that the co-repressor BCL6, when knocked out in CD8+ T cells, promotes glycolysis, decreases exhaustion, and leads to better effector

function and tumor control in transplantable tumor models. The studies utilize BCL6-floxed mice and granzyme-cre, CD4-cre, OX49-cre, Mb1-cre, and RAG mice to demonstrate that BCL6 in CD8+ T cells but not in CD4+, NK, or B cells play a role in antitumor immunity. The study also uses a BCL6 inhibitor as a pharmacologic treatment of cancer. Human data confirm that patients with tumors with low BCL6 have higher survival. This conclusively demonstrates that BCL6 restricts CD8+ tumor immunity and inhibition of BCL6 could be a viable therapeutic option. This advances the field of cancer immune therapy and CD8+ T cell differentiation.

The main point that BCL6 restricts CD8+ T cell antitumor activity is strongly supported by the data. Additional experiments to show this are not needed.

The mechanism of BCL6 as a repressor of GLUT3 is shown by luciferase reporter assays, RNAseq studies of BCL6-deficient CD8+ T cells, GLUT3 qrtPCR measurements, and glycolysis assays. These support a role for BCL6 in regulating GLUT3 but do not demonstrate that this activity underlies the increased anti-tumor activity of BCL6-deficient CD8+ T cells. Improved glycolysis is known to increase CD8+ antitumor activity, but is this the sole mechanism of BCL6 activity in CD8+ T cells? An experiment in which GLUT3 is knocked out in BCL6-KO CD8+ T cells might show if regulating GLUT3 is required for BCL6's restraining of CD8+ antitumor activity.

We thank the reviewer for these comments. Breeding the BCL6 CD8 T cell-specific KO mice with GLUT3 KO mice might help address this concern, but the time and cost of cross-breeding may be too much. The underlying mechanism of how BCL6 regulates glycolysis has been uncovered in previous studies. BCL6 has been reported to directly repress genes involved in the glycolysis pathway, including *Slc2a1*, *Slc2a3*, *Pkm2*, *Hk2* and HIF-1 α (PMID: 25194422, PMID: 36542153). The molecular mechanism in this case may not depend solely on GLUT3 derepression; therefore, we have tempered our conclusions accordingly in the manuscript. In addition, there are other functional or potential roles of BCL6 in CD8 T cells that do not rely on glycolysis. As we already mentioned in the manuscript, BCL6 antagonizes BLIMP1 to promote an intratumor stem/progenitor-like CD8⁺ T cell (T_{prog} cell) population [1]. BCL6 controls T cell non-follicular positioning by silencing *Ccr7*, *Ccr6*, *S1pr1*, and *Psgl1* [2], suggesting a deficiency of BCL6 may change the distribution of activated CD8 T cells, but whether this could affect tumor immunity needs further investigation.

The possibility that BCL6 promotes exhaustion in CD8⁺ T cells in tumors is supported by findings that BCL6-deficient intratumoral CD8⁺ T cells demonstrate slightly higher effector activity and lower expression of exhaustion/dysfunction markers (PD1, TIM3, CD39). This could be due to smaller tumor sizes and changed tumor microenvironment. Whether BCL6 regulates transcription factors thought to be involved in exhaustion is not known, but the RNAseq data suggest that it does not. The relationship between glycolysis/GLUT3 and exhaustion is also not demonstrated.

It would be good to look at whether overexpression of GLUT3 in CD8+ T cells reverses exhaustion.

Criboli E et al. [3] reported that enforcing GLUT3 expression in CD8 T cells results in significantly lower expression of exhaustion markers such as PD-1, Lag-3, and Tim-3 and drives higher glucose uptake. We added this point to the discussion as well to establish the link between GLUT3/glycolysis and exhaustion.

The studies with the inhibitor show therapeutic potential and suggests that BCL6 in effector CD8+ T cells is restraining activity. The phenotype seems weaker than the genetic deletion. This might suggest a role for BCL6 during priming. Perhaps giving the inhibitor earlier would test if BCL6 plays a role in priming.

A premature start of the treatment will cause tumor initiation failure in the recipients, hence we introduced FX1 treatment on day 4 after MC38 inoculation, and i.p. injected the FX1 (25 mg/kg) every day till day 10. Even though mice treated with FX1 showed smaller tumors (a), it is probably because of the efficacy of the early dose of drugs. We didn't find any differences in the proportion of CD44+, PD1+, or CD69+ CD8 T cells within tumors between the solvent and FX1-treated groups (b), suggesting that the priming or activation of CD8 T cells is not significantly different whether BCL6 is eliminated or not. The weaker tumor-resistant phenotype observed when using a

BCL6 inhibitor compared to genetic deletion of BCL6 expression in activated CD8 T cells is because we wait for the tumors to grow to approximately 200 mm³, which is considered more convincing for evaluating drug efficacy.

Figure for reviewers #1

No other issues to address.

Reviewer #2 (Comments to the Authors (Required)):

In this study, Luan et al. demonstrate that BCL6 suppresses the anti-tumor activity of CD8⁺ T cells. Using a series of T cell specific Cre drivers, the authors show that BCL6 knockout in CD8⁺ T cells enhances tumor control in a subcutaneous model across multiple cell lines. This improved control appears to be mediated by increased effector function and reduced exhaustion of CD8⁺ T cells. Notably, the GzmB-Cre model used deletes BCL6 upon activation, which may explain why the authors did not observe

differences in the TCF1⁺ population previously observed. Transcriptomic analysis of BCL6-deficient CD8⁺ T cells suggested metabolic alterations, including increased glycolysis, suggesting a mechanistic basis for enhanced cellular fitness. The authors convincingly demonstrated that glycolysis is improved in BCL6 deficient T cells. Finally, the authors demonstrate that pharmacologic inhibition of BCL6 after tumor initiation can improve anti-tumor immunity.

Overall, the manuscript is clearly written and presents a well-executed study on the role of BCL6 in the anti-tumor response of CD8⁺ T cells. While the novelty of targeting BCL6 to enhance anti-tumor responses has been somewhat diminished by the Sun et al. publication (Science Immunology 2023), the authors take a distinct approach and provide additional insight identifying altered metabolism as a mechanism contributing to increase fitness in the BCL6 knockout cells. Although their identification of GLUT3 as a target to improve T cell fitness has already been confirmed in CAR-T cell mouse models. Given the differences between the previous model and the total knockout in transgenic T cells previously published, it would be informative to understand the expression of Bcl6 and its loss in the author's system. The authors are to be commended for the rigor of their work, including clear reporting of statistics, mouse numbers, and replicates.

Comments to the authors:

1. As the model used a Cre controlled under the GzmB promoter rather than an E8I-Cre, BCL6 is lost only upon activation and its knockout is not restricted to CD8 T cells. Although it appears that BCL6 is not expressed highly in naïve CD8s it would be

good to know if BCL6 is expressed by any CD8s in the tumor microenvironment, or if the promoter essentially functions as a total knockout in the CD8 compartment. Additionally, although GzmB is associated more with CD8s, CD4 T cells can express GzmB when activated. Staining T cells for BCL6 from the tumor microenvironment would also assess the degree to which the CD4 compartment loses BCL6 expression under this promoter.

We thank the reviewer for these comments. We measured BCL6 levels on tumor-infiltrating CD8 T cells on DPI 10 and DPI 21. The results (Figure S2I) demonstrated that our model indeed knocked out BCL6 expression on activated CD8 T cells rather than activated CD4 T cells.

2. In Figure 4B, the tumors in Rag1 knockout mice appear to have the same growth kinetics as the WT mice. I found this interesting as it has been reported that the adaptive immune system contributes some at baseline to control of MC38 tumors (e.g. Efremova et al. Nat. Com 2018). I am curious if the authors normally see no difference in growth between WT and Rag1 KO mice with their inoculating dose and if they do see a difference when taking the tumor out longer?

Data from one of our other projects, MC38 tumor inoculations in Rag1 KO mice on DPI 16 has slightly higher tumor volume than that in C57BL/6 mice ($p=0.613$). However, when combined with these data or if we look later, on DPI 19, both MC38 (a) or B16F10 (b) tumor growth curves seem quite comparable between Rag1 KO mice

and WT mice.

Figure for Reviewer #2.

3. In Figure 2, I hesitate to define a T cell as exhausted solely based on co-expression of PD-1 and TIM-3, or dysfunction based on PD-1 and CD39 expression. These markers are associated with exhaustion but are not the definition. It could strengthen the interpretation to include an analysis showing the proportion of T cells that are single-, double-, or triple-positive for these inhibitory receptors, providing a better view of the exhaustion spectrum within the population.

Truly, the definition of T cell exhaustion or dysfunction is a complicated subject, as we discussed in the manuscript. We also found that a consistently lower percentage of tumor-infiltrating dysfunctional or exhausted T cells in the *Bcl6^{fl/fl}Gzmb-cre* mice than that in their controls when we gated on CD39+CD69+PD1+ CD8 T cells (Figure 2F) and CD39+CD69+Tim3+PD1+CD44+ CD8 T cells (Figure 2G) within tumors on DPI 20.

4. In Figure 3B, it may just be my copy, but some gene names appear clipped off

ensure that all are readable in the final image.

Thanks for the comment. We enlarged the text size of gene names in Figure 3B.

Reviewer #3 (Comments to the Authors (Required)):

In this study, Luan et al. demonstrate a role of BCL6 in the antitumor CD8+ T cell response in flank transplant tumor models, and show that BCL6 knockout enhances effector functions and decreases the proportion of exhausted T cells. They also report that BCL6 represses GLUT3 by directly binding to the Slc2a3 locus, and that BCL6 knockout thus increases GLUT3 and enhance glycolysis in these T cells, which may underlie their enhanced function and decreased exhaustion. Finally, the authors use an inhibitor of BCL6, Fx1, to show that this checkpoint can be pharmacologically targeted to enhance antitumor T cell responses in vivo. The study is mostly well-executed and the manuscript clearly written. However, the results are largely derivative in that the role of BCL6 in CD8 T cells has been studied in a number of publications in both viral and cancer models, and a role for BCL6 in repressing glycolysis via repression of Slc2a3 in CD4 helper T cells (Th1) has been previously reported (Oestreich, KJ, et al., Nature Immunology, 15(10):957-964, 2014). Nevertheless, the present manuscript provides an important angle on cancer immunotherapy and would still be valuable to the field, but there are a few critical points that must be addressed prior to publication.

1) The human TCGA analysis correlating BCL6 expression (bulk RNA) to overall survival is problematic and the conclusions drawn (that high BCL6 expression indicates infiltration by dysfunctional T cells) are not supported by these data. This analysis should either be removed or replaced by one that is more nuanced, with IHC/IF staining of T cells, or analysis of single cell datasets that can enable specific enumeration of BCL6-expressing T cells.

We thank the reviewer for these comments. We swapped the Figure 1A and Figure S1D upon the reviewer's request.

2) The authors mention some prior work on the role of BCL6 in T cells, but review of this prior literature is incomplete. The authors should more thoroughly integrate the prior literature into their introduction and discussion, and provide a nuanced discussion of how their findings fit into this broader picture. Specifically, these two papers should be discussed more thoroughly and with comparison to the present manuscript's results: Liu, Z, et al., J Immunol (2019) 203 (2): 323-327, Sun, Q, et al., and Science Immunology (2023) 88 (8). While these studies come to slightly different conclusions regarding the role of BCL6 in generating versus maintaining memory T cells, they agree on the importance of this gene in driving expression of TCF1 and supporting the memory progenitor transcriptional program while antagonizing expression of effector genes.

It would seem that Luan et al., in the present study, have generated results consistent with these prior studies, but have focused only on the enhancement of the effector

response early on, while avoiding the question of memory responses, persistence of BCL6^{-/-} T cells, and longer-term immunity. While it may be outside the scope of the present study to perform memory response/recall experiments, the authors should discuss these nuances in detail. For example, while BCL6 knockout may enhance the early effector response by promoting terminal effector differentiation, this may be at the expense of the longer-term memory response, something that is not accurately recapitulated in short-term cell line transplant models.

In response to the reviewers' concerns about how to fit our work into the broader literatures, even though how BCL6 regulates CD8 T cell memory response is not the scope of this study, we conducted a deeper discussion among these nuances. We also added the finding of the paper [4] that the reviewer mentioned above into the introduction.

Specific Comments

1)

54-55 "Compared to effector or memory T cells, the category of exhausted T cells encompasses a broad definition,"

A little misleading, memory and effector T cells also come in many types/definitions. I

suggest eliminating the "compared to effector or memory T cells" clause.

As requested by the reviewer, we deleted this attributive adverbial from the sentence.

2)

75 "BCL6 is also critical for maintaining and generating memory CD8 T cells [21]"

The results of this study [21] are somewhat contradictory to the messaging of the present manuscript-that BCL6 deletion can INCREASE the quality and function of the antitumor T cell response. If generation and maintenance of memory CD8 T cells is impaired by BCL6 knockout, then this would seem a poor strategy to increase antitumor immunity. However, this cited paper has the major caveat of using germline BCL6 knockout mice, which could affect hematopoiesis and all stages of T cell (and other immune cell) development, potentially resulting in phenotypically/functionally different naïve CD8 T cells used in the transfer studies. In addition, there is another paper that reported that conditional BCL6 knockout in CD8 T cells is required for the generation, but not maintenance, of memory CD8 T cells (J Immunol (2019) 203 (2): 323-327), although another study reported that BCL6 deficiency impairs the persistence, but not generation, of memory progenitor CD8 T cells (Sun, Q, et al., and Science Immunology (2023) 88 (8)). Despite these somewhat conflicting results, both of these studies align in their report that BCL6 positively regulates the expression of the Tcf7 gene to support the memory progenitor state.

The authors should discuss these prior findings in much more detail and how their work fits into this broader literature:

This concern seems similar to the comment #2) mentioned above, we added more discussion to explain how our work fits into the findings around the field.

Specifically, the authors report that Bcl6-flox/flox; Gzmb-Cre CD8 T cells in the MC38 model do not show decreased TCF1 expression, which would conflict with the results of the prior studies mentioned above. However, looking closely at Figure 2C, it appears that there actually is a (probably) significant decrease in TCF1 expression in these CD8 T cells at day 8, but not 20. The authors should report statistical significance here, and even if not significant, mention the trend and rationalize their results with the prior literature. MFI is also a rather crude way of plotting these data. The authors should also show percent or total. Is it possible that BCL6 knockout is skewing the balance of memory progenitor to effector ratio towards more short-lived effector cells, and thus early on in the response there are more TCF1 low effector T cells? Whereas later in the response, the short-lived effector T cells have contracted (consistent with the Sun, Q, et al. study above reporting that these Bcl6^{-/-} effectors have impaired persistence), leaving only the memory/memory progenitor cells with high levels of TCF1? It would also be helpful to assess the degree/efficiency of BCL6 knockout at the 8-day and 20-day timepoints in the model, as there may be selection against BCL6 knockout later in the response, given reported defects in persistence, resulting in enrichment of BCL6^{+/+} or BCL6^{+/-} CD8 T cells that failed to undergo complete recombination. As mentioned in another comment, it is important to validate

that the Cre-recombinase system is actually resulting in efficient deletion of BCL6 in CD8 T cells at the timepoints analyzed.

In Figure 2C, we provided the p value ($p=0.0814$) of TCF1(MFI) between *Bcl6*^{fl/fl} Gzmb-cre and *Bcl6*^{fl/fl} on DPI 8. As shown below, the proportion of TCF1+CD44+ CD8 T cells within tumors was not significantly different between *Bcl6*^{fl/fl} Gzmb-cre and *Bcl6*^{fl/fl} mice. And the validation of our Cre-driven BCL6 knock-out models were presented in Figure S2I.

Figure for Reviewer #3.

3)

75-78 "The expression level of BCL6 exhibits a strong positive correlation with the number of central memory CD8 T cells (TCM) upon antigen stimulation and contributes to antigen-specific TCM secondary expansion"

Expression level in what context? This sentence confusing. Are the authors saying that the expression level of BCL6 in central memory T cells is strongly correlated to other markers associated with central memory? Or the expression level of BCL6 in T cells prior to activation correlates with their subsequent development into central memory? Or that the expression level of BCL6 in T cells following antigen stimulation correlates with their subsequent development into central memory T cells? Seems like probably the last, but the wording could be clearer.

We rephased the sentence as requested by the reviewer.

4)

156-163 "We found that tumor cases with lower levels of Bcl6 mRNA had favorable overall survival rates in all types of cancer (Fig. 1A) and this phenotype was confirmed in colon adenocarcinoma (COAD, with $p=0.049$), pancreatic adenocarcinoma (PAAD, with $p=0.021$) and, possibly, rectum adenocarcinoma (READ, no significant difference but trend was shown) (Fig. S1A-161 C). Moreover, tumor-infiltrating CD8 T cell levels and Bcl6 expression showed a positive correlation in COAD, PAAD and READ samples (Fig. S1D-F), suggesting that BCL6 promotes CD8 T cell proliferation within tumors while suppressing their killing function."

This analysis is fraught and the conclusions counterintuitive. There are numerous

publications demonstrating that levels of CD8 T cell infiltration within solid tumors like the ones analyzed here are associated with better survival. It is also impossible in this analysis to determine if BCL6 expression is coming from CD8 T cells or tumor/stromal/other immune cells. To support the authors' hypothesis convincingly, it would be important to show that BCL6 level in tumors is driven mostly by CD8 T cells, and/or that BCL6+ CD8 T cells (via IF/IHC, flow, or scRNA-seq) correlates with survival or some other feature strongly associated with survival. Otherwise, the argument is very weak. One possible explanation is that tumor intrinsic expression of BCL6 is associated with a more malignant phenotype, despite these tumors having a slightly elevated level of T cell infiltration (relatively weak Rho), or these T cells are excluded at the edges of the tumor.

We deleted the hypothesis here, and only described the observation.

5)

190-191 "Moreover, the CD39+PD1+ and CD69+PD1+ dysfunctional CD8 T cells"

Why are CD69+PD1+ CD8s being called dysfunctional? This could also just indicate a resident effector phenotype or a recently activated phenotype.

By gating CD39+CD69+PD1+ CD8 T cells (Figure 2F) and CD39+CD69+Tim3+PD1+CD44+ CD8 T cells (Figure 2G) within tumors on DPI 20, we presented better results to show lower levels of dysfunctional/exhausted T cells

when BCL6 was eliminated from activated CD8 T cells than their controls.

6)

193-197 "These data suggest that as the size of inoculated tumors increase the function of CD8 T cells transforms from effector to exhaustion or dysfunction, and that the elimination of BCL6 in activated CD8 T cells can reverse this process to some extent, allowing them to exert their killing effector for a longer time."

This is a strange way to interpret the data. Do the authors actually show that the size of the tumor is associated with exhaustion phenotypes? And is that really the point? A better way to say this would be that in the face of an actively growing tumor and persistent antigen stimulation, the infiltrating T cells become progressively exhausted/dysfunctional. But this is a well-appreciated concept, and doesn't really need to be said. It suffices to say that BCL6 deletion reduces the acquisition of features of exhaustion in tumor-infiltrating CD8 T cells in the model. The authors should also be precise here with wording related to reversal versus prevention of exhaustion. Given the early time points assessed in the model, the experimental results support a role of BCL6 knockout in preventing exhaustion at the time point analyzed, not reversing established exhaustion.

We removed the wording of reversing exhaustion and rephrased the sentence as requested by the reviewer.

7)

235 "We first confirmed that BCL6 expression was strongly induced upon T cell activation (Fig. 4A)."

What about BCL6 levels in T cells treated with Fx1? Is this inhibitor actually specific and on-target? It should be very straightforward to assess the effects of Fx1 treatment on BCL6 expression levels in T cells in vitro, and would also be nice to see in the in vivo experiments at the doses used in the tumor growth studies.

As shown below, we detected a repressed level of BCL6 expression on tumor-infiltrating activated CD8 T cells after 5 doses of FX1.

Figure for Reviewer #3.

8)

303-304 "Targeting exhausted T cells to restore their effector function is an independent, non-mutually exclusive approach clearly worth further investigations."

Immune checkpoint blockade also targets exhausted T cells, albeit through a different mechanism, so the first clause of this sentence is not accurate. Are the authors drawing the distinction that ICB targets progenitor exhausted T cells while BCL6 inhibition/Fx1 is specifically targeting terminally exhausted T cells? The distinction is not clear, and the authors have not clearly shown that BCL6 inhibition is acting in any way on terminally exhausted T cells to rescue their function. Rather, BCL6 inhibition may be preventing the onset of exhaustion specifically in the proliferating effector population. Alternatively, it is possible that BCL6 knockout is resulting in the early contraction/apoptosis of exhausted T cells, based on the impaired persistence of BCL6 knockout effector CD8 T cells reported in Sun, Q, et al., and therefore there could be fewer exhausted T cells left (another reason that looking at overall numbers/magnitude of the T cell response and these different populations between experimental groups is really important). Importantly, the authors start treatment with Fx1 at day 10, which is likely close to the peak of priming and proliferative expansion of effector T cells in the model, and not a time point when many T cells would already be exhausted, making it impossible to disentangle effects on exhausted T cell generation versus existing exhausted T cells.

This raises an important point that the authors should address in their Discussion. Does BCL6 inhibition prevent exhaustion or rescue it, and if only the former, it is unlikely to work effectively in established tumors due to the majority of infiltrating T cells already being dysfunctional. This also raises a major limitation of flank transplant models-the kinetics are too rapid to easily separate treatment effects across distinct phases of the T cell response. It would also be worth entertaining the possibility that BCL6 inhibition could synergize with ICB, if the above mechanism proves true, as ICB would enable proliferative expansion of early-exhausted T cells with effector functions from progenitor exhausted T cells, and BCL6 inhibition may prolong their proliferative and effector function through maintaining glycolysis and proliferation. To be clear, these nuances should be entertained in the Discussion and no additional experiments are being requested to resolve these specific questions.

Firstly, we replaced “reversed” to “prevent” or “retard” to avoid raising the confusion about any role of BCL6 in reversing exhaustion. Secondly, we added some lines to discuss the potential behavior of BCL6 in combination therapies with ICB.

9)

Given the well-established role of BCL6 in the cell cycle and the fact that rapidly proliferating cells are associated with glycolytic metabolism, the authors should

assess the effects of BCL6 knockout/inhibition on T cell proliferation using EDU or other proliferation assays, and by quantifying/reporting the magnitude of the T cell response in the in vivo models. Is proliferation enhanced with BCL6 knockout, and does this result in greater numbers of primed T cells in LNs/spleen/or blood and increased tumor-infiltrating CD8 T cells?

BCL6 knockout leading to more T cell proliferation probably could not be possible.

Knocking-out BCL6 should lead to slower proliferation of cells based on the following well-established studies. BCL6 directly represses *TP53*, *CDKN1A*, *ATR* and *CHEK1* [5-9] to facilitate proliferation and survival. Additionally, BCL6 downregulates cell cycle inhibitors PTEN, p21 and p27 to advance the cell cycle transition from G1 to S phase [6, 10], using BCL6 inhibitor (FX1 or 79-6) to reactivate BCL6 target genes results in G1 arrest [11-13].

10)

The authors should show that their BCL6 knockout CD8 T cells actually have loss of BCL6 protein, by Western, flow, or other methods. It would be important to include the controls without Cre lines, and also CD4 T cells. This is quite important to establishing that the model is actually functioning as intended and the phenotypes are indeed due to BCL6 knockout, especially in the later timepoints where there may be selection against BCL6 knockout CD8 T cells due to impaired persistence.

As shown in Figure S2I, we confirmed the knocked-out levels of BCL6 protein in tumor-infiltrating activated CD8 T cells from our *Bcl6^{fl/fl}* Gzmb-cre mice compared with their controls.

References

1. Sun Q, Cai D, Liu D, Zhao X, Li R, Xu W, et al. BCL6 promotes a stem-like CD8(+) T cell program in cancer via antagonizing BLIMP1. *Sci Immunol*. 2023; 8: eadh1306.
2. Qi H. T follicular helper cells in space-time. *Nat Rev Immunol*. 2016; 16: 612-25.
3. Cribioli E, Giordano Attianese GMP, Ginefra P, Signorino-Gelo A, Vuillefroy de Silly R, Vannini N, et al. Enforcing GLUT3 expression in CD8(+) T cells improves fitness and tumor control by promoting glucose uptake and energy storage. *Front Immunol*. 2022; 13: 976628.
4. Liu Z, Guo Y, Tang S, Zhou L, Huang C, Cao Y, et al. Cutting Edge: Transcription Factor BCL6 Is Required for the Generation, but Not Maintenance, of Memory CD8(+) T Cells in Acute Viral Infection. *J Immunol*. 2019; 203: 323-7.
5. Phan RT, Dalla-Favera R. The BCL6 proto-oncogene suppresses p53 expression in germinal-centre B cells. *Nature*. 2004; 432: 635-9.
6. Phan RT, Saito M, Basso K, Niu H, Dalla-Favera R. BCL6 interacts with the transcription factor Miz-1 to suppress the cyclin-dependent kinase inhibitor p21 and cell cycle arrest in germinal center B cells. *Nat Immunol*. 2005; 6: 1054-60.
7. Ranuncolo SM, Polo JM, Dierov J, Singer M, Kuo T, Grealley J, et al. Bcl-6 mediates the germinal center B cell phenotype and lymphomagenesis through transcriptional repression of the DNA-damage sensor ATR. *Nat Immunol*. 2007; 8:

705-14.

8. Ranuncolo SM, Wang L, Polo JM, Dell'Oso T, Dierov J, Gaymes TJ, et al. BCL6-mediated attenuation of DNA damage sensing triggers growth arrest and senescence through a p53-dependent pathway in a cell context-dependent manner. *J Biol Chem*. 2008; 283: 22565-72.
9. Parekh S, Prive G, Melnick A. Therapeutic targeting of the BCL6 oncogene for diffuse large B-cell lymphomas. *Leuk Lymphoma*. 2008; 49: 874-82.
10. Ci W, Polo JM, Cerchietti L, Shaknovich R, Wang L, Yang SN, et al. The BCL6 transcriptional program features repression of multiple oncogenes in primary B cells and is deregulated in DLBCL. *Blood*. 2009; 113: 5536-48.
11. Cerchietti LC, Ghetu AF, Zhu X, Da Silva GF, Zhong S, Matthews M, et al. A small-molecule inhibitor of BCL6 kills DLBCL cells in vitro and in vivo. *Cancer Cell*. 2010; 17: 400-11.
12. Sultan M, Nearing JT, Brown JM, Huynh TT, Cruickshank BM, Lamoureaux E, et al. An in vivo genome-wide shRNA screen identifies BCL6 as a targetable biomarker of paclitaxel resistance in breast cancer. *Mol Oncol*. 2021; 15: 2046-64.
13. Ishikawa C, Mori N. FX1, a BCL6 inhibitor, reactivates BCL6 target genes and suppresses HTLV-1-infected T cells. *Invest New Drugs*. 2022; 40: 245-54.

September 15, 2025

RE: Life Science Alliance Manuscript #LSA-2025-03335R

Prof. David Nemazee
Scripps Research Institute
Department of Immunology
10550 Torrey Pines Road, IMM-29
La Jolla, CA 92037

Dear Dr. Nemazee,

Thank you for submitting your revised manuscript entitled "Loss of Bcl6 promotes anti-tumor immunity by activating glycolysis to rescue CD8 T cell function". As you will see, reviewers were broadly supportive of this work upon reviewing the changes in place.

Reviewers 2 and 3 both expressed concerns related to TCF1 expression in BCL6 KO T cells. We concur with Reviewer 2 that the data showing percent TCF1 positive cells included for reviewers must be included in Figure 2, alongside the current panel 2C showing MFI. In this regard the results and discussion (lines 273-4) should be revised to acknowledge potentially reduced expression.

Several concerns of Reviewer 3 remain outstanding, and we appreciate their thorough consideration of these issues and prior literature. A suitably revised manuscript must either temper claims based on gene expression in Figure 1A, or include public scRNA seq datasets to report on BCL6 expression specifically in T cells. We concur that the current observations do not directly address effects of BCL6 on memory T cells in anti-tumor immunity, which must be made more explicit in the results and discussion. Finally, we encourage inclusion of data on normalized T cell counts, per Reviewer 3 point 9, although generating new data on T cell proliferation is not required. Additional changes beyond those mentioned above are left to your discretion.

We will assess these changes without further reviewer input, and will be happy to publish your paper in Life Science Alliance pending resolution of the above requests and final revisions necessary to meet our formatting guidelines.

- Please add the X and Bluesky handles of your host institute/organization, as well as your own and/or one of the authors in our system.
- Please consult our manuscript preparation guidelines <https://www.life-science-alliance.org/manuscript-prep> and make sure your manuscript sections are in the correct order.
- The "Data Availability" section should be placed after the Materials & Methods section. Please consult our guidelines at <https://www.life-science-alliance.org/manuscript-prep#format>
- Please add your main, supplementary figure, and table legends to the main manuscript text after the references section and remove them from the figures.
- Please be sure that the authorship listing and order is correct.
- While the current title is acceptable, we suggest this alternative:
"Loss of Bcl6 promotes anti-tumor immunity by activating glycolysis to rescue CD8 T cell function."

A. FINAL FILES:

B. MANUSCRIPT ORGANIZATION AND FORMATTING:

Thank you for your attention to these final processing requirements. Please revise and format the manuscript and upload materials as soon as you are able.

Sincerely,

Reviewer #1 (Comments to the Authors (Required)):

The revised version has addressed my main issues and presents a high-quality set of studies with unbiased and appropriate conclusions and discussion of the data in the context of other findings. I do not have any other issues with the manuscript.

Reviewer #2 (Comments to the Authors (Required)):

In this study, Luan et al. demonstrate that BCL6 suppresses the antitumor activity of CD8 T cells by increasing exhaustion and reducing effector function. They observed this phenotype across multiple tumor cell lines in a subcutaneous model using a system of Cre-specific drivers to KO BCL6 in different lymphocyte populations. Their GzmB-Cre primarily targets activated CD8 T cells, explaining the effects observed. The authors then performed RNA-seq and followed up observations of altered metabolism with experiments demonstrating that GLUT3 is repressed by BCL6, impacting T cell glycolysis. They performed Cut&Run and a luciferase assay to demonstrate that BCL6 suppressed GLUT3 expression. I agree with Review 1 that a KO of GLUT3 would have to be performed to prove that this is the whole mechanism of action in altering metabolism and rescuing the T cell phenotype, but the authors altered the language accordingly. Finally, the authors demonstrate that pharmacologic

inhibition of BCL6 after tumor initiation can improve antitumor immunity.

Overall, the manuscript is clearly written and presents a well-executed study on the role of BCL6 in the antitumor response of CD8⁺ T cells. The authors reined in the strength of their language after the first review to appropriately match their presented data and added some controls to better validate their BCL6 KO and inhibition by FX1.

Comment to the authors:

1. The authors included flow staining to show that there was a decrease in Bcl6 in the CD8⁺ but not CD4⁺ population (Fig S2I). The CD8⁺ T cells are additionally gated on PD1⁺ for an antigen-experienced population, which makes sense given that Cre expression is driven off GzmB, but making the comparison to just the CD4⁺ population is a little unfair. Gating on a marker of antigen experience would make this comparison fairer. While this flow data is not super striking, they demonstrated clearly that the antigen-experienced CD8 population had reduced Bcl6 expression. This, plus the OX40-Cre mice showing no differences, supports that it's a CD8 T cell-driven phenotype.
2. I agree with Reviewer 3 that the plotting of MFI in Figure 2C is a bit more crude than percent of parent. The authors may want to swap this graph in place of the MFI currently in that slot.
3. Thank you for including additional gating on multiple markers of exhaustion. I think this supports the conclusion that Bcl6 KO reduces the degree of exhaustion.

Reviewer #3 (Comments to the Authors (Required)):

In this study, Luan et al. demonstrate a role of BCL6 in the antitumor CD8⁺ T cell response in flank transplant tumor models, and show that BCL6 knockout enhances effector functions and decreases the proportion of exhausted T cells. They also report that BCL6 represses GLUT3 by directly binding to the Slc2a3 locus, and that BCL6 knockout thus increases GLUT3 and enhance glycolysis in these T cells, which may underlie their enhanced function and decreased exhaustion. Finally, the authors use an inhibitor of BCL6, Fx1, to show that this checkpoint can be pharmacologically targeted to enhance antitumor T cell responses in vivo. The study is mostly well-executed and the manuscript clearly written. However, the results are largely derivative in that the role of BCL6 in CD8 T cells has been studied in a number of publications in both viral and cancer models, and a role for BCL6 in repressing glycolysis via repression of Slc2a3 in CD4 helper T cells (Th1) has been previously reported (Oestreich, KJ, et al., *Nature Immunology*, 15(10):957-964, 2014). Nevertheless, the present manuscript provides an important angle on cancer immunotherapy and would still be valuable to the field, but there are a few critical points that must be addressed prior to publication.

1) The human TCGA analysis correlating BCL6 expression (bulk RNA) to overall survival is problematic and the conclusions drawn (that high BCL6 expression indicates infiltration by dysfunctional T cells) are not supported by these data. This analysis should either be removed or replaced by one that is more nuanced, with IHC/IF staining of T cells, or analysis of single cell datasets that can enable specific enumeration of BCL6-expressing T cells.

"We thank the reviewer for these comments. We swapped the Figure 1A and Figure S1D upon the reviewer's request."

This does not address my original concern that BCL6 expression may be coming from other cells other than CD8 T cells, as both these figures draw from bulk expression analysis that is underpowered to make convincing conclusions about expression from CD8 T cells specifically. Because these results are hypothesis generating and used to introduce the mechanistic experiments that follow, it is not essential that the authors perform IHC/IF of BCL6 and CD8a/b on human samples, but I strongly recommend performing this analysis in one of the many existing published single cell RNA-sequencing datasets to be able to definitively associate BCL6 expression with CD8 T cells. In the absence of these data, the authors should at the very least include a strong caveat about the limitations of their TCGA analysis.

2) The authors mention some prior work on the role of BCL6 in T cells, but review of this prior literature is incomplete. The authors should more thoroughly integrate the prior literature into their introduction and discussion, and provide a nuanced discussion of how their findings fit into this broader picture. Specifically, these two papers should be discussed more thoroughly and with comparison to the present manuscript's results: Liu, Z, et al., *J Immunol* (2019) 203 (2): 323-327, Sun, Q, et al., and *Science Immunology* (2023) 88 (8). While these studies come to slightly different conclusions regarding the role of BCL6 in generating versus maintaining memory T cells, they agree on the importance of this gene in driving expression of TCF1 and supporting the memory progenitor transcriptional program while antagonizing expression of effector genes. It would seem that Luan et al., in the present study, have generated results consistent with these prior studies, but have focused only on the enhancement of the effector response early on, while avoiding the question of memory responses, persistence of BCL6^{-/-} T cells, and longer-term immunity. While it may be outside the scope of the present study to perform memory response/recall experiments, the authors should discuss these nuances in detail. For example, while BCL6 knockout may enhance the early effector response by promoting terminal effector differentiation, this may be at the expense of the longer-term memory response, something that is not accurately recapitulated in short-term cell line transplant models.

"In response to the reviewers' concerns about how to fit our work into the broader literatures, even though how BCL6 regulates CD8 T cell memory response is not the scope of this study, we conducted a deeper discussion among these nuances. We also added the finding of the paper [4] that the reviewer mentioned above into the introduction."

These additions are appreciated. While the manuscript is not focused on memory responses per se, it does explore TCF1, a critical mediator of memory progenitor T cells in the early T cell response and later memory. Furthermore, the claims that BCL6 inhibition represents a promising immunotherapeutic strategy are limited by the fact that the model systems only enable analysis of the T cell response within a very short time frame, and it remains possible that the improved responses observed could be transient and at the expense of longer-term, durable responses. This is a critical point of discussion that should be added.

Also, Lines 271-286 in the Discussion are a bit confusing and would improve with some clearer language and restructuring. For example, the mention of "fluctuations" would be clearer if the authors stated something like, "Expression levels of TCF1 and BCL6 in T cells do not covary consistently, with naïve T cells expressing TCF1 but not BCL6, and TOX+ progenitor-like T cells expressing high levels of both". A follow-up sentence about the implications of this would help to clarify why this is relevant to the authors' interpretation of their results.

Specific Comments

1)

54-55 "Compared to effector or memory T cells, the category of exhausted T cells encompasses a broad definition,"

A little misleading, memory and effector T cells also come in many types/definitions. I suggest eliminating the "compared to effector or memory T cells" clause.

"As requested by the reviewer, we deleted this attributive adverbial from the sentence."

Okay

2)

75 "BCL6 is also critical for maintaining and generating memory CD8 T cells [21]"

The results of this study [21] are somewhat contradictory to the messaging of the present manuscript—that BCL6 deletion can INCREASE the quality and function of the antitumor T cell response. If generation and maintenance of memory CD8 T cells is impaired by BCL6 knockout, then this would seem a poor strategy to increase antitumor immunity. However, this cited paper has the major caveat of using germline BCL6 knockout mice, which could affect hematopoiesis and all stages of T cell (and other immune cell) development, potentially resulting in phenotypically/functionally different naïve CD8 T cells used in the transfer studies. In addition, there is another paper that reported that conditional BCL6 knockout in CD8 T cells is required for the generation, but not maintenance, of memory CD8 T cells (*J Immunol* (2019) 203 (2): 323-327), although another study reported that BCL6 deficiency impairs the persistence, but not generation, of memory progenitor CD8 T cells (Sun, Q, et al., and *Science Immunology* (2023) 88 (8)). Despite these somewhat conflicting results, both of these studies align in their report that BCL6 positively regulates the expression of the *Tcf7* gene to support the memory progenitor state.

The authors should discuss these prior findings in much more detail and how their work fits into this broader literature:

"This concern seems similar to the comment #2) mentioned above, we added more discussion to explain how our work fits into the findings around the field."

The response to main comment #2) above does not address the apparent contradiction of BCL6 inhibition in T cells promoting antitumor immunity yet impairing the generation and maintenance of memory T cells. Again, this is a major caveat to the authors' claims that BCL6 inhibition represents a promising immunotherapy strategy and should be thoroughly discussed. I insist that this is a critical discussion point, because there is substantial prior literature regarding the role of BCL6 in memory T cells, and memory T cell responses are considered critical for durable antitumor T cell responses. The authors cited the above paper that found that "BCL6 is critical for maintaining and generating memory CD8 T cells", yet have failed to reconcile the apparent contradiction to clinical translation of their findings: how can an approach that impairs lasting CD8 T cell memory hold promise for durable antitumor immunity in human cancer? The discussion must address this.

Specifically, the authors report that *Bcl6*-flox/flox; *Gzmb*-Cre CD8 T cells in the MC38 model do not show decreased TCF1 expression, which would conflict with the results of the prior studies mentioned above. However, looking closely at Figure 2C, it appears that there actually is a (probably) significant decrease in TCF1 expression in these CD8 T cells at day 8, but not 20. The authors should report statistical significance here, and even if not significant, mention the trend and rationalize their results with the prior literature. MFI is also a rather crude way of plotting these data. The authors should also show percent or total CD8+/CD44+ TILs that are TCF1+. Is it possible that BCL6 knockout is skewing the balance of memory progenitor to effector ratio towards more short-lived effector cells, and thus early on in the response there are more TCF1 low effector T cells? Whereas later in the response, the short-lived effector T cells have contracted (consistent with the Sun, Q, et al. study above reporting that these *Bcl6*^{-/-} effectors have impaired persistence), leaving only the memory/memory progenitor cells with high levels of TCF1? It would also be helpful to assess the degree/efficiency of BCL6 knockout at the 8-day and 20-day timepoints in

the model, as there may be selection against BCL6 knockout later in the response, given reported defects in persistence, resulting in enrichment of BCL6^{+/+} or BCL6^{+/-} CD8 T cells that failed to undergo complete recombination. As mentioned in another comment, it is important to validate that the Cre-recombinase system is actually resulting in efficient deletion of BCL6 in CD8 T cells at the timepoints analyzed.

"In Figure 2C, we provided the p value ($p=0.0814$) of TCF1 (MFI) between Bcl6fl/fl Gzmb-cre and Bcl6fl/fl on DPI 8. As shown below, the proportion of TCF1⁺CD44⁺ CD8 T cells within tumors was not significantly different between Bcl6fl/fl Gzmb-cre and Bcl6fl/fl mice. And the validation of our Cre-driven BCL6 knock-out models were presented in Figure S21."

I appreciate that this change has now been made. A p-value of 0.08 is borderline significant and implies that BCL6 knockout does indeed reduce TCF1 expression. It certainly does not indicate that BCL6 knockout has no effect on TCF1 expression levels and memory/progenitor differentiation, as implied by what is currently written in the Results:

188-192 There was no difference found in TCF1 expression when comparing activated CD44⁺ CD8 T cells of Bcl6fl/fl Gzmb-cre and Bcl6fl/fl mice (Fig. 2C), suggesting that knocking out BCL6 during CD8 T cell activation may not have influenced their stemness and memory differentiation.

A lack of statistical power to prove a trend significant is by no means proof of the negative (no difference). Quite the opposite, these data suggest there may indeed be an effect of BCL6 knockout on TCF1 expression levels. Given the importance of TCF1 expression levels to the nature of the T cell response (memory progenitor versus short-lived effector versus memory, etc.), and the aforementioned points about prior literature on the role of BCL6 in T cell memory development (which is not limited to late-stage memory but begins early after priming), it is very important that this result is definitively shown and contextualized with the prior literature. The authors should repeat this experiment and determine if the trend is the same and perform a combined statistical analysis and make a definitive statement about how BCL6 knockout results in lower TCF1 expression (or not) early in the effector response. Combined with the TCF⁺ fraction analysis, this should help determine if BCL6 is influencing TCF1 expression levels per cell or TCF1⁺ cell population dynamics due to an influence on differentiation. Related to differentiation, the authors' inclusion of percentage of T cells that are CD44⁺/TCF1⁺ is appreciated, but this approach is not ideal, given that inclusion of all T cells in this percentage can be skewed by uninvolved (e.g., blood/lymphatic vessel-associated and naïve T cells). The authors should gate first on antigen-experienced (CD44⁺) CD8 T cells and then assess percentage of TCF1⁺ cells within this population. Inclusion of other markers of progenitor versus short-lived effector CD8 T cells should also be included (e.g., KLRG1, GZMB, CD127, etc.).

Regarding validation of Cre-driven BCL6 knockout, this was not shown in the previous manuscript (as implied by the wording in the authors' response). The new Figure S2 subpanels are appreciated, though they are missing letter labels (i.e., H, I). Also, in Figure S21 (presumably, the last figure on the bottom right), the Y-axis shows percent of PD1⁺/CD39⁺ CD8 T cells-the authors should show BCL6 expression levels for all antigen-experienced (CD44⁺) CD8 T cells instead, and also gate on BCL6⁺ versus BCL6⁻ using the appropriate control samples and FMOs, and convincingly show that a significant fraction of the antigen-experienced compartment is indeed knockout for BCL6. The authors may also use TCF1 and GZMB positivity for gating, as it is expected that knockout is specific to the TCF1⁻ population, or the GZMB⁺ effector population where Gzmb-Cre is expected to be expressed.

3)

75-78 "The expression level of BCL6 exhibits a strong positive correlation with the number of central memory CD8 T cells (TCM) upon antigen stimulation and contributes to antigen-specific TCM secondary expansion"

Expression level in what context? This sentence confusing. Are the authors saying that the expression level of BCL6 in central memory T cells is strongly correlated to other markers associated with central memory? Or the expression level of BCL6 in T cells prior to activation correlates with their subsequent development into central memory? Or that the expression level of BCL6 in T cells following antigen stimulation correlates with their subsequent development into central memory T cells? Seems like probably the last, but the wording could be clearer.

"We rephrased the sentence as requested by the reviewer."

Where is the rephrased sentence? The authors should paste in the changes they have made in response to reviewer comments in the rebuttal so the reviewers can adequately evaluate changes.

4)

156-163 "We found that tumor cases with lower levels of Bcl6 mRNA had favorable overall survival rates in all types of cancer (Fig. 1A) and this phenotype was confirmed in colon adenocarcinoma (COAD, with $p=0.049$), pancreatic adenocarcinoma (PAAD, with $p=0.021$) and, possibly, rectum adenocarcinoma (READ, no significant difference but trend was shown) (Fig. S1A-

161 C). Moreover, tumor-infiltrating CD8 T cell levels and Bcl6 expression showed a positive correlation in COAD, PAAD and READ samples (Fig. S1D-F), suggesting that BCL6 promotes CD8 T cell proliferation within tumors while suppressing their killing function."

This analysis is fraught and the conclusions counterintuitive. There are numerous publications demonstrating that levels of CD8 T cell infiltration within solid tumors like the ones analyzed here are associated with better survival. It is also impossible in this analysis to determine if BCL6 expression is coming from CD8 T cells or tumor/stromal/other immune cells. To support the authors' hypothesis convincingly, it would be important to show that BCL6 level in tumors is driven mostly by CD8 T cells, and/or that BCL6+ CD8 T cells (via IF/IHC, flow, or scRNA-seq) correlates with survival or some other feature strongly associated with survival. Otherwise, the argument is very weak. One possible explanation is that tumor intrinsic expression of BCL6 is associated with a more malignant phenotype, despite these tumors having a slightly elevated level of T cell infiltration (relatively weak Rho), or these T cells are excluded at the edges of the tumor.

"We deleted the hypothesis here, and only described the observation."

This is inadequate. What is the purpose of presenting an observation that is based on a questionable analysis without further explanation or context? Removal of the hypothesis only makes this result even more confusing to the reader.

As with my earlier comment on using single cell RNA-seq to specifically show BCL6 expression levels within CD8 T cells, the authors must perform this analysis robustly in a way that specifically isolates BCL6 expression level within the CD8 T cell compartment.

5)

190-191 "Moreover, the CD39+PD1+ and CD69+PD1+ dysfunctional CD8 T cells"

Why are CD69+PD1+ CD8s being called dysfunctional? This could also just indicate a resident effector phenotype or a recently activated phenotype.

"By gating CD39+CD69+PD1+ CD8 T cells (Figure 2F) and CD39+CD69+Tim3+PD1+CD44+ CD8 T cells (Figure 2G) within tumors on DPI 20, we presented better results to show lower levels of dysfunctional/exhausted T cells 21 when BCL6 was eliminated from activated CD8 T cells than their controls."

The authors should provide a more complete and less biased exhaustion marker co-expression analysis, as is standard in the field for analysis of exhausted T cells (the more markers co-expressed is accepted to be associated with greater degrees of exhaustion, see seminal T cell exhaustion papers from the Wherry lab for reference). That is, showing percentages of the antigen-experienced (CD44+) CD8 T cell population that express one, two, three, four, etc. exhaustion markers (depending on how many exhaustion markers were included in the flow panels). It should also be noted that CD69 is not an exhaustion marker, even though it can be associated with exhausted T cells-rather, it is a marker of tissue residence. PD-1, CD39, and TIM3 are appropriate exhaustion markers.

6)

193-197 "These data suggest that as the size of inoculated tumors increase the function of CD8 T cells transforms from effector to exhaustion or dysfunction, and that the elimination of BCL6 in activated CD8 T cells can reverse this process to some extent, allowing them to exert their killing effector for a longer time."

This is a strange way to interpret the data. Do the authors actually show that the size of the tumor is associated with exhaustion phenotypes? And is that really the point? A better way to say this would be that in the face of an actively growing tumor and persistent antigen stimulation, the infiltrating T cells become progressively exhausted/dysfunctional. But this is a well-appreciated concept, and doesn't really need to be said. It suffices to say that BCL6 deletion reduces the acquisition of features of exhaustion in tumor-infiltrating CD8 T cells in the model. The authors should also be precise here with wording related to reversal versus prevention of exhaustion. Given the early time points assessed in the model, the experimental results support a role of BCL6 knockout in preventing exhaustion at the time point analyzed, not reversing established exhaustion.

"We removed the wording of reversing exhaustion and rephrased the sentence as requested by the reviewer."

199-201 These data suggest that the elimination of BCL6 in activated CD8 T cells can prevent the acquisition of features of T cell exhaustion within tumors, allowing them to exert their killing effector for a longer time.

The first part of this sentence is a reasonable conclusion, but the second is more speculative and should be rephrased to say, "and may allow them to exert their cytotoxic effector functions for a longer time."

7)

235 "We first confirmed that BCL6 expression was strongly induced upon T cell activation (Fig. 4A)."

What about BCL6 levels in T cells treated with Fx1? Is this inhibitor actually specific and on-target? It should be very straightforward to assess the effects of Fx1 treatment on BCL6 expression levels in T cells in vitro, and would also be nice to see in the in vivo experiments at the doses used in the tumor growth studies.

"As shown below, we detected a repressed level of BCL6 expression on tumor-infiltrating activated CD8 T cells after 5 doses of FX1."

This is appreciated but should also be shown for more broadly (not just PD1+) antigen-specific (CD44+) T cells. The authors should also show percent BCL6+ fraction, not just MFI, because again, MFI can reflect per T cell BCL6 expression levels OR shifts in the population of T cells that do and do not express BCL6 (see previous comments).

8)

303-304 "Targeting exhausted T cells to restore their effector function is an independent, non-mutually exclusive approach clearly worth further investigations."

Immune checkpoint blockade also targets exhausted T cells, albeit through a different mechanism, so the first clause of this sentence is not accurate. Are the authors drawing the distinction that ICB targets progenitor exhausted T cells while BCL6 inhibition/Fx1 is specifically targeting terminally exhausted T cells? The distinction is not clear, and the authors have not clearly shown that BCL6 inhibition is acting in any way on terminally exhausted T cells to rescue their function. Rather, BCL6 inhibition may be preventing the onset of exhaustion specifically in the proliferating effector population. Alternatively, it is possible that BCL6 knockout is resulting in the early contraction/apoptosis of exhausted T cells, based on the impaired persistence of BCL6 knockout effector CD8 T cells reported in Sun, Q, et al., and therefore there could be fewer exhausted T cells left (another reason that looking at overall numbers/magnitude of the T cell response and these different populations between experimental groups is really important). Importantly, the authors start treatment with Fx1 at day 10, which is likely close to the peak of priming and proliferative expansion of effector T cells in the model, and not a time point when many T cells would already be exhausted, making it impossible to disentangle effects on exhausted T cell generation versus existing exhausted T cells.

This raises an important point that the authors should address in their Discussion. Does BCL6 inhibition prevent exhaustion or rescue it, and if only the former, it is unlikely to work effectively in established tumors due to the majority of infiltrating T cells already being dysfunctional. This also raises a major limitation of flank transplant models-the kinetics are too rapid to easily separate treatment effects across distinct phases of the T cell response. It would also be worth entertaining the possibility that BCL6 inhibition could synergize with ICB, if the above mechanism proves true, as ICB would enable proliferative expansion of early-exhausted T cells with effector functions from progenitor exhausted T cells, and BCL6 inhibition may prolong their proliferative and effector function through maintaining glycolysis and proliferation. To be clear, these nuances should be entertained in the Discussion and no additional experiments are being requested to resolve these specific questions.

"Firstly, we replaced "reversed" to "prevent" or "retard" to avoid raising the confusion about any role of BCL6 in reversing exhaustion. Secondly, we added some lines to discuss the potential behavior of BCL6 in combination therapies with ICB."

Okay

9)

Given the well-established role of BCL6 in the cell cycle and the fact that rapidly proliferating cells are associated with glycolytic metabolism, the authors should assess the effects of BCL6 knockout/inhibition on T cell proliferation using EDU or other proliferation assays, and by quantifying/reporting the magnitude of the T cell response in the in vivo models. Is proliferation enhanced with BCL6 knockout, and does this result in greater numbers of primed T cells in LNs/spleen/or blood and increased tumor-infiltrating CD8 T cells?

"BCL6 knockout leading to more T cell proliferation probably could not be possible. Knocking-out BCL6 should lead to slower proliferation of cells based on the following well-established studies. BCL6 directly represses TP53, CDKN1A, ATR and CHEK1 [5-9] to facilitate proliferation and survival. Additionally, BCL6 downregulates cell cycle inhibitors PTEN, p21 and p27 to advance the cell cycle transition from G1 to S phase [6, 10], using BCL6 inhibitor (FX1 or 79-6) to reactivate BCL6 target genes results in G1 arrest [11-13]."

This is not necessarily true. BCL6 has context-specific effects on cell cycle depending on cell type and state, and its acute effects on cell cycle are distinct from its effects on T cell differentiation states, as reported in the literature. It is also challenging to decouple the acute effects of BCL6 knockout/inhibition on cell cycle from the effects on progenitor state and differentiation

towards exhaustion and terminal differentiation, which initially can rapidly proliferate but ultimately lose proliferative capacity over time. This is to say, the authors' explanation here is inadequate and I insist it is critical that they examine the effects of BCL6 knockout on CD8 T cell proliferation. It would be simple enough for the authors to reexamine their existing flow cytometry data to quantify the overall magnitude/numbers of antigen-specific T cells in the different experimental groups, if they do not have EDU/Ki67/etc. data or do not want to perform additional experiments to quantify proliferation specifically. Proliferation may be increased or decreased, it is unclear, but either way, it is an important analysis that should be included.

10)

The authors should show that their BCL6 knockout CD8 T cells actually have loss of BCL6 protein, by Western, flow, or other methods. It would be important to include the controls without Cre lines, and also CD4 T cells. This is quite important to establishing that the model is actually functioning as intended and the phenotypes are indeed due to BCL6 knockout, especially in the later timepoints where there may be selection against BCL6 knockout CD8 T cells due to impaired persistence.

"As shown in Figure S2I, we confirmed the knocked-out levels of BCL6 protein in tumor-infiltrating activated CD8 T cells from our *Bcl6^{fl/fl} Gzmb-cre* mice compared with their controls."

See response to the last part of point 2).

The following is a point by point reply from the authors addressing the most recent critique.

We thank the comments of 3 reviewers again.

Followed by reviewer #2's suggestion, we included the calculation of TCF1+CD44+ CD8 T cells on day 8 post-MC38 tumor inoculation in Figure 2D.

We appreciate reviewer 3's thorough consideration of our results and prior literature.

We summarize our responses to the newly added comments of reviewer 3 below:

Main point 1 combined with specific point 4: We include a new Figure 1A showing BCL6 levels in CD8 T cells specifically from a public scRNAseq database.

Main point 2 combined with specific point 2: We reiterate our assertion to Reviewer 3 again that T memory responses are not within the scope of this study. Moreover, we could not present any further data to show memory T cell behaviors in our model to establish a deeper discussion based on evidence. Hence, we expressed more explicitly the insignificant trend of TCF1 level changes in BCL6 deficient activated CD8 T cells in the results. It is unreasonable to be forced to fit our results that showed no significant differences to a published paper in a different model. We have already added multiple points in the discussion to let our work fit into a broader field upon the request of reviewer 3 in the last revision, and our interpretation of the "*BCL6 in a narrow cohort of cells in a limited developmental window, leaving other CD8 T cells untouched*" suggest a mechanism already included in the discussion. Although this issue was a concern of Reviewer 3, we point out that the other reviewers did not seem

to agree.

Specific points 1 and 8 were resolved.

Specific point 3: The rephrased sentence is located at lines 69-72 of the revised manuscript, which we provided last time. It can be tracked by reference [22] easily.

Specific point 5: We already provided the 4 different types of gating to present the exhausted or dysfunctional T cells, which has dispelled the concern of reviewer 2, who had a similar concern last time.

Specific point 6: We changed “allowing” to “may allow” in the sentence upon the newly added request of reviewer 3.

Specific point 7: This data “Figure for Reviewer #3” shown in the rebuttal letter at “specific comment 7”. And it has addressed the question “What about BCL6 levels in T cells treated with Fx1?” which reviewer 3 raised last time, we did not see the necessity of reanalyzing this data.

Specific point 9: We added the results of no significant differences between percentage and number of MC38 tumor-infiltrating TCF1+CD44+ CD8 T cells on DPI 8 at Figure 2D. For better fluency of writing and reading, we posted the calculation of CD44+ CD8 T cell number between Bcl6f/f and Bcl6f/f Gzmb-cre mice after 8-day tumor stimulation below to address the assumption of reviewer 3 rather than showing it into the manuscript.

Figure for reviewer #3.

Specific point 10: In the last revision, we added the data to Figure S2 and labeled it with “1”. We quoted reviewer 2’s new comments 1 *“While this flow data is not super striking, they demonstrated clearly that the antigen-experienced CD8 population had reduced Bcl6 expression.”*, supporting that our results have met the requirement.

October 15, 2025

RE: Life Science Alliance Manuscript #LSA-2025-03335RR

Prof. David Nemazee
Scripps Research Institute
Department of Immunology
10550 Torrey Pines Road, IMM-29
La Jolla, CA 92037

Dear Dr. Nemazee,

Thank you for submitting your Research Article entitled "Loss of Bcl6 promotes anti-tumor immunity by activating glycolysis to rescue CD8 T cell function". We appreciate your inclusion of a rebuttal to the remaining points by Reviewers 2 and 3. It is a pleasure to let you know that your manuscript is now accepted for publication in Life Science Alliance. Congratulations on this interesting work.

DISTRIBUTION OF MATERIALS:

Again, congratulations on a very nice paper. I hope you found the review process to be constructive and are pleased with how the manuscript was handled editorially. We look forward to future exciting submissions from your lab.

Sincerely,
